# CryoNet.Refine: A One-step Diffusion Model for Rapid Refinement of Structural Models with Cryo-EM Density Map Restraints

**Fuyao Huang**[2,†]    **Xiaozhu Yu**[2,†]    **Kui Xu**[3,*]    **Qiangfeng Cliff Zhang**[1,*]

[1] State Key Laboratory of Membrane Biology, Beijing Frontier Research Center of Biological Structures,
Tsinghua-Peking Joint Center for Life Sciences, School of Life Sciences,
Tsinghua University, Beijing, China
[2] State Key Laboratory of Membrane Biology–Membrane Structure and Artificial Intelligence Biology Branch,
Hangzhou, China
[3] State Key Laboratory of Membrane Biology,
Beijing Tsinghua Institute for Frontier Interdisciplinary Innovation, Beijing, China

[†] These authors contributed equally to this work.
[*] Correspondence: `xukui.2016@tsinghua.org.cn`, `qczhang@tsinghua.edu.cn`

## Abstract

High-resolution structure determination by cryo-electron microscopy (cryo-EM) requires the accurate fitting of an atomic model into an experimental density map. Traditional refinement pipelines like *Phenix.real_space_refine* and Rosetta are computationally expensive, demand extensive manual tuning, and present a significant bottleneck for researchers. We present *CryoNet.Refine*, an end-to-end, deep learning framework that automates and accelerates molecular structure refinement. Our approach utilizes a one-step diffusion model that integrates a density-aware loss function with robust stereochemical restraints, enabling it to rapidly optimize a structure against the experimental data. *CryoNet.Refine* stands as a unified and versatile solution capable of refining not only protein complexes but also DNA/RNA-protein complexes. In benchmarks against *Phenix.real_space_refine*, *CryoNet.Refine* consistently yields substantial improvements in both model–map correlation and overall model geometric metrics. By offering a scalable, automated, and powerful alternative, *CryoNet. Refine* is poised to become an essential tool for next-generation cryo-EM structure refinement. Web server: `https://cryonet.ai/refine/`; Source code: `https://github.com/kuixu/cryonet.refine`.

## 1 Introduction

Cryo-electron microscopy (cryo-EM) has emerged as a revolutionary technique in structural biology, enabling the determination of macromolecular structures, including numerous crucial biological complexes, at unprecedented resolution (Kühlbrandt, 2014; Nogales & Eva, 2016). The typical cryo-EM workflow involves several sequential stages: sample preparation, electron micrograph acquisition, particle picking, three-dimensional (3D) reconstruction, atomic model building, and final structure refinement. Despite these advancements, a persistent challenge in cryo-EM remains the inherent low signal-to-noise ratio (SNR) and the pervasive conformational dynamics of biological samples. These factors often lead to low-resolution cryo-EM density maps from 3D reconstruction, and even high-resolution maps frequently exhibit low-resolution densities at peripheral and/or flexible regions. Such limitations severely compromise the effectiveness of atomic model building tools, including traditional approaches like *Phenix.map_to_model* (Afonine et al., 2018a), *A2-Net* (Xu et al., 2019), *DeepTracer*(Pfab et al., 2021), *CryoNet*(`https://cryonet.ai`), and even more recent breakthroughs like *ModelAngelo*(Jamali et al., 2024). This can result in fragmented or

incomplete structures, incorrect identification of amino acid or nucleic acid types, and in extreme cases, the inability to complete atomic model building, often necessitating integration with orthogonal data sources like RNA-seq for structure discovery(Wang et al., 2024; 2025).

Due to the limitations mentioned above, atomic model refinement becomes a crucial step to follow once an initial atomic model is built. This phase meticulously adjusts the atomic coordinates of each amino acid or nucleic acid within the model, typically employing both automated and manual interactive tools. Automated tools, exemplified by *Phenix.real_space_refine* (Afonine et al., 2018b), are highly versatile, capable of refining not only protein structures but also DNA and RNA. It incorporates a comprehensive library of structural restraints, including secondary structure, rotamer, and Ramachandran plot. By iteratively optimizing the stucture through simulated annealing and sampling from vast conformational spaces, it seeks a subset of models that best fit the cryo-EM density map, ultimately yielding a structure with excellent geometric metrics and high model-map correlation coefficients. Concurrently, interactive tools like *Coot*(Emsley & Cowtan, 2004) provide structural biologists with user-friendly interfaces to visualize poor geometry for convenient manual adjustments. Coot further allows users to dynamically adjust the weights of local density regions with unbalanced quality, offering effective real-time constraints during manual refinement. While highly effective, both automated and manual methods often require "case-by-case" parameter tuning by experts. The process of cryo-EM refinement is in urgent need of a flexible, robust and fully automated method . Leveraging the powerful capabilities of modern AI approaches thus holds immense prospect for developing superior cryo-EM refinement tools.

The landscape of AI in structural biology has been dramatically reshaped by recent advancements in generative models, particularly diffusion models, which excel in protein generation, design, and structure prediction. Breakthroughs like *AlphaFold3*(Abramson et al., 2024), *RFDiffusion*(Watson et al., 2023), and *Chroma*(Ingraham et al., 2023) exemplify their exceptional ability to generate diverse and high-quality structures for various biomolecules, including proteins, DNA, and RNA complexes. Beyond single-chain protein prediction, where models like *AlphaFold2*(Jumper et al., 2021), *RoseTTAFold* (Baek et al., 2021), and *ESMFold*(Lin et al., 2023) have achieved remarkable accuracy, diffusion-based methods are extending to protein-protein interactions, protein-small molecule complexes, and DNA/RNA complexes (Baek et al., 2021; 2023; Wohlwend et al., 2025; Passaro et al., 2025). Notably, *AlphaFold3* has moved beyond predefined fixed bond lengths and angles from *AlphaFold2*, by learning these geometric constraints directly from PDB data using diffusion models. This includes accurately capturing geometry features like the planarity of benzene rings in amino acids. However, a significant limitation of these generative methods is that while they produce geometrically plausible structures, they often exhibit suboptimal performance on various detailed geometric metrics and, crucially, do not natively support refinement under the direct restraints of experimental data, such as cryo-EM density maps. Applying the robust generative ability of diffusion models to cryo-EM data-restraints refinement thus offers a transformative pathway to substantially improve structure quality and overcome the reliance on laborious, parameter-heavy traditional or manual refinement workflows.

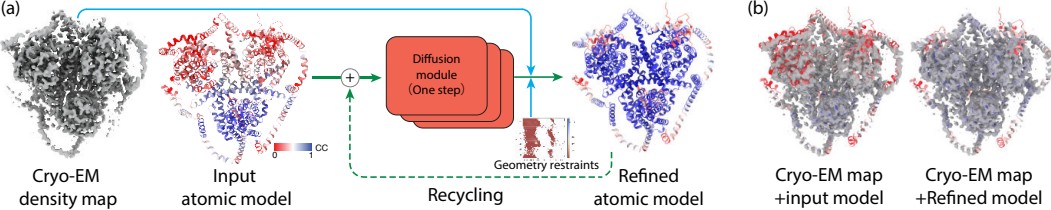

Figure 1: **(a)** Workflow of *CryoNet.Refine*. Atomic models are colored by model–map correlation coefficients (CC), with blue indicating high CC and red indicating low CC. **(b)** The input atomic model, refined atomic model within cryo-EM density map.

To address these limitations, we introduce *CryoNet.Refine*, a novel one-step diffusion model designed for cryo-EM atomic model refinement Figure 1.**(a)**. *CryoNet.Refine* pioneers the integration of advanced atomic generation principles, inspired by the capabilities seen in *AlphaFold3*, into a comprehensive refinement framework. Taking a cryo-EM density map and an atomic structure as input, Atom encoder first extracts intricate features from the structure to be refined. Concurrently, Sequence embedder encodes atomic type information derived from the input molecular sequence.The Pairformer module follows the *Boltz-2*(Passaro et al., 2025) perform cross-attention between the

atom and the sequence embeddings. These representations are then fed into our one-step diffusion module, which iteratively generates the refined atomic structure. The refinement process is guided by novel designed density loss and a set of geometry losses, facilitating iterative optimization of the atomic model. *CryoNet.Refine* achieves the first successful atomic structure refinement under the direct constraints of experimental cryo-EM density maps and the guidance of standard geometry metrics within a neural network framework (Figure 1.(**b**)). Experimental benchmarkds demonstrates significant improvements over *Phenix.real_space_refine* on both protein-protein and DNA/RNA-protein complexes, showcasing its exceptional performance and broad applicability. Our work presents four main contributions:

(1) We propose the first AI-based method for cryo-EM atomic model refinement, leveraging a deep neural network-based one-step diffusion module.
(2) We develop a novel parameter-free and differentiable density generator that can produce simulated density maps from the generated atomic model. This innovation enables us to design an effective density loss, marking the first time that density map correlation can be directly utilized as a loss function to guide neural network training.
(3) We introduce a set of differentiable geometry loss functions specifically tailored for guiding the generation of geometrically plausible macromolecular structures, which also offer valuable guidance for protein design and protein structure prediction models.
(4) We conduct extensive evaluation of *CryoNet.Refine* against *Phenix.real_space_refine* on cryo-EM datasets, showing a marked improvement in structural quality across various metrics.

## 2 RELATED WORK

In this section, we review the state-of-the-art methods in cryo-EM atomic model refinement and structure modeling, highlighting both traditional and emerging AI-driven approaches.

**Cryo-EM atomic model refinement.** Cryo-electron microscopy (cryo-EM) has revolutionized structural biology, enabling the determination of macromolecular structures at near-atomic resolution. However, the initial cryo-EM density maps often require subsequent atomic model refinement to achieve biologically meaningful and geometrically accurate structures. Traditionally, atomic model refinement methods can be broadly categorized into automated and manual approaches. Automated tools, such as *Phenix.real_space_refine*(Afonine et al., 2018b), *Rosetta*(Wang et al., 2016), and *ISOLDE*(Croll, 2018), leverage various geometric restraints (e.g., bond lengths, angles, dihedral angles) and force fields to drive atomic models to fit to density while maintaining stereochemical integrity. While these methods can be powerful and yield highly refined structures and, with expert "case-by-case" parameter tuning, their extensive parameter sets and specialized workflows often present a steep learning curve for non-expert users. In parallel, manual refinement tools like *Coot*(Emsley & Cowtan, 2004)offer powerful interactive visualization capabilities, allowing researchers to meticulously adjust individual amino acid residues or local regions directly within the density map. These tools provide unparalleled control and flexibility, enabling highly precise adjustments. Nevertheless, manual refinement is notoriously labor-intensive, time-consuming, and similarly demands significant expert knowledge, making it a bottleneck for high-throughput structure determination.

**Protein Structure Refinement.** More recently, the field has seen a surge of interest in artificial intelligence (AI)-driven methods for Protein Structure Refinement. Approaches such as *DeepAcc-Net*(Hiranuma et al., 2021), *GNNRefine*(Jing & Xu, 2021), and *AtomRefine*(Wu et al., 2023) employ neural networks, typically 3D convolution neural networks and Graph Neural Networks (GNNs), to learn intricate geometric features of protein backbones and side-chains. These methods aim to predict corrections or refine atomic positions based on learned structural patterns. A key characteristic of these existing AI-based refinement techniques is their primary reliance on structural learning, often from large databases of known protein structures. Consequently, the refined structures are fundamentally predictions of geometrically plausible conformations. A critical limitation, however, is the general absence of direct integration with experimental cryo-EM density maps during the differentiable optimization process. This disconnect means that the final predicted structures, while potentially ideal in terms of stereochemical geometry, frequently do not optimally match experimental data. Currently, there is a notable gap in the literature for neural network-based methods that support differentiable refinement under the direct constraint of cryo-EM experimental data.

**Diffusion models for structural generation.** The advent of diffusion models has marked a significant paradigm shift in generative AI, demonstrating remarkable capabilities in complex data gener-

ation, including protein structure prediction and design. Diffusion models, exemplified by architectures like *RFDiffusion*(Watson et al., 2023) for de novo protein design and *AlphaFold3*(Abramson et al., 2024) for biomolecular interaction structure prediction and, excel at learning atom distributions and generating diverse, high-quality biomolecular structures. Their success in protein structure generation is largely attributed to their ability to capture global and local structural features, implicitly enforcing geometric characteristics like bond lengths and angles, thus generating models with high stereochemical quality. These models have shown exceptional proficiency in maintaining geometric fidelity in generated structures. Leveraging the strong structural priors and generative capabilities of diffusion models could offer a novel avenue for cryo-EM structure refinement, particularly if their powerful learning frameworks could be adapted to integrate and be driven by experimental cryo-EM density information in a differentiable manner.

# 3 METHODS

## 3.1 CRYONET.REFINE FRAMEWORK

*CryoNet.Refine* is an end-to-end deep learning framework designed for the atomic model refinement of macromolecules directly against experimental cryo-EM density maps Figure 2.**(a)**. The refinement process begins by taking an experimental cryo-EM density map ($d_0$) and an initial atomic structure ($x_0$) as input. Atom encoder first processes the input structure to extract pairwise features ($z$) for each atom. Concurrently, Sequence embedder encodes atomic type information ($s$) derived from the input molecular sequence. These encoded representations ($z$,$s$) along with the initial atomic structure ($x_0$) are then fed into our one-step Diffusion Module to generate a refined atomic structure ($x_1$). Subsequently, the density generator takes the refined structure ($x_i$) as input to produce a simulated density map ($d_i$). A designed density loss ($\mathcal{L}_{\text{den}}$) is then calculated by comparing this simulated map ($d_1$) with the input map ($d_0$). Simultaneously, a Geometry loss ($\mathcal{L}_{\text{geo}}$) is computed based on the refined structure ($x_i$). These two losses are then weighted and summed to form the total loss ($\mathcal{L}$), which is backpropagated to update the parameters of the diffusion module. Crucially, this workflow represents a test-time optimization strategy rather than a static inference pass. For each specific case, the network undergoes this iterative training-and-optimization loop to fine-tune the parameters of the diffusion module, thereby customizing the atomic structure against the experimental density map and geometric constraints. This process constitutes one refinement cycle. After $n$ such cycles, the final refined atomic model, $x_n$, is obtained. Our network was first initialized with the parameters of *Boltz-2*(Passaro et al., 2025), a PyTorch implementation of *AlphaFold3*, followed by training with a composite loss function including two components: one measuring experimental data fidelity and the other introducing geometric restraints. The network was trained until convergence, yielding the generated atomic structures that are in high agreement with the experimental data while satisfying geometric constraints.

**One-step diffusion module.** The atomic structure generator is designed as a one-step diffusion model. Unlike the diffusion module in *AlphaFold3* that requires hundreds of sampling steps, one-step diffusion models represent a major development in generative AI, leveraging techniques like Knowledge Distillation(Meng et al., 2022) and Consistency Models(Song et al., 2023) (as explored in (Wu et al., 2025)) to compress the generation process and reduce computational bottlenecks. We have observed that the one-step diffusion module possesses a key advantage: it can easily incorporate features from experimental data and geometric restraints, into the generation process. This one-step design effectively and efficiently respects the given guidance, generating accurate results that align tightly with the restraints. Figure 2.**(b)** shows the refining atomic model trajectory colored with model-map correlation coefficients.
Concretely, the refining process is performed by training the network under a **preconditioned parameterization** following (Karras et al., 2022), and finally producing the refined coordinates $\hat{\mathbf{x}}$ as:

$$\hat{\mathbf{x}} = c_{\text{skip}}(\sigma)\,\mathbf{x}_0 + c_{\text{out}}(\sigma)\,\mathcal{F}_\theta\Big(c_{\text{in}}(\sigma)\mathbf{x}_0,\, c_{\text{noise}}(\sigma),\, \mathcal{C}\Big), \qquad (1)$$

where $c_{\text{skip}}$, $c_{\text{out}}$ and $c_{\text{out}}$ denote coefficients in the preconditioned forward module(Appendix C) and $\mathcal{F}_\theta$ is a parameterized neural network in it. $\mathcal{C}$ denotes conditioned features derived from encoded structural features $s$ and $z$. This preconditioned update ensures that the one-step refinement preserves the theoretical properties of multi-step diffusion while collapsing the stochastic process into a deterministic, single-step prediction.
**Restraints of density map and structure geometry.** *CryoNet.Refine* employs a density generator

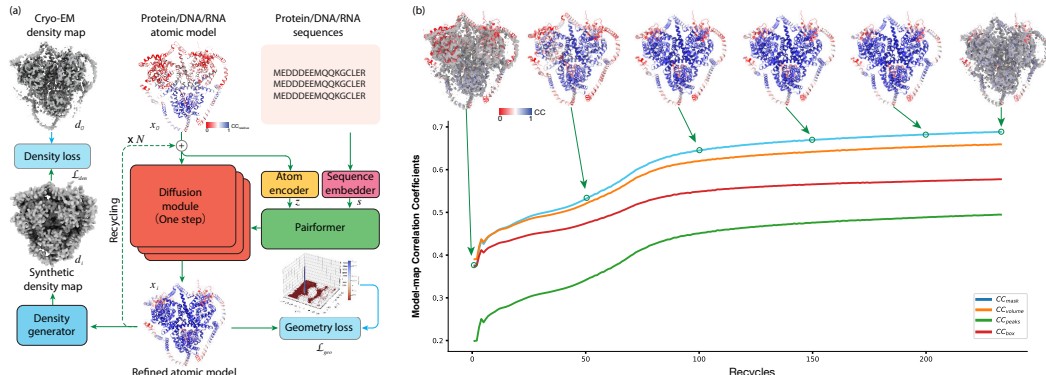

Figure 2: **(a)** Overview of the *CryoNet.Refine* framework. The *CryoNet.Refine* Network consists of five modules: Atom encoder, Sequence embedder, Pairformer module, Diffusion module, and Density generator. The input atomic model is first processed by the encoders, and the resulting features are fed into a one-step diffusion module to generate an initial refined atomic model. Subsequently, the Density generator creates a synthetic density map, which is used to compute density loss against the input density map and geometry loss based on geometry restraints. These losses are then backpropagated to optimize the diffusion module, while the atomic model is further refined through multiple recycle steps until convergence. **(b)** Model-map correlation coefficient trajectory over 234 recycling *CryoNet.Refine* iterations on the structure of the human concentrative nucleoside transporter CNT3((Bank)). The input density map is EMD-0775, the input atomic model is predicted by *AlphaFold3*.

and two types of loss functions to leverage these restraints. The first type of loss is a **density loss**, derived by compute the correlation between the experimental cryo-EM map and the synthetic map generated from refined atomic model by the density generator (Section 3.2.1). Another one is a set of geometry loss terms imposes stereochemical restraints—such as those derived from Ramachandran plot, rotamer, and bond angle distributions—to ensure standard structure geometry (Section 3.2.2). Thus the network is trained using a recycling strategy that reuses refined outputs as inputs for subsequent passes, progressively optimizing the two loss functions for density map agreement and geometry metrics. Together, these novel design choices enable CryoNet.Refine to achieve rapid, generalizable, and experimentally consistent refinement by fully respecting the restraints derived from both the cryo-EM density map and the structure geometry.

## 3.2 LOSS FUNCTIONS

The loss function of *CryoNet.Refine* consists of density loss and geometry loss, defined as follows:

$$
\begin{aligned}
\mathcal{L} &= \gamma_{\text{den}} \cdot \mathcal{L}_{\text{den}} + \mathcal{L}_{\text{geo}}, \\
\mathcal{L}_{\text{geo}} &= \gamma_{\text{rama}} \cdot \mathcal{L}_{\text{rama}} + \gamma_{\text{rot}} \cdot \mathcal{L}_{\text{rot}} + \gamma_{\text{angle}} \cdot \mathcal{L}_{\text{angle}} + \gamma_{\text{C}_\beta} \cdot \mathcal{L}_{\text{C}_\beta} + \gamma_{\text{viol}} \cdot \mathcal{L}_{\text{viol}},
\end{aligned}
\tag{2}
$$

where $\gamma$ is the weight of each loss. See the detailed definition of these functions in Section 3.2.1, Section 3.2.2, and Appendix E. To the best of our knowledge, *CryoNet.Refine* is the first to formulate a differentiable implementation to compute density loss, ramachandran loss, rotamer loss and $C_\beta$ loss. Although these biological implications have been well-established and recognized as indispensable for protein structure prediction and atomic model building, prior methodologies have not integrated similar constraints into their nerual network based refinement processes.

### 3.2.1 DENSITY LOSS

We compute density loss in two steps: (i) generate a synthetic density map for the refined atomic model; (ii) compute cosine similarity between the input and the synthetic density map to yield the loss term. Both these two steps must be implemented to be fully differentiable. Here we briefly describe how the differentiable density generator works and the formula for density loss function, with more details discussed in Appendix D.

Our current density generator is not a neural network but a physics-based simulator. Given the grid points $\vec{m}$ and all-atom coordinates of the structure model $\vec{x}$ with $N$ atoms, we get the density values

$\hat{\rho}$ by constructing Gaussian spheres centering at each atom position:

$$\hat{\boldsymbol{\rho}}(\vec{\boldsymbol{m}}, \vec{\mathbf{x}}) = \sum_{i=1}^{N} w_i e^{-k|\vec{\boldsymbol{m}} - \vec{\mathbf{x}}_i|^2} \tag{3}$$

where $w_i$ represents the atomic number of the $i^{th}$ atom, $\vec{\mathbf{x}}$ is the position vector of the $i^{th}$ atom, and $k$ is determined by the synthetic density map resolution $res$ as well as the voxel size $v$ (both specified as identical to the input experimental map):

$$k = 8 \cdot res / (\pi \cdot v) \tag{4}$$

Based on the synthetic density values $\hat{\boldsymbol{\rho}}(\vec{\boldsymbol{m}}, \vec{\mathbf{x}})$ and those retrieved from the input density map $\boldsymbol{\rho}(\vec{\boldsymbol{m}})$, we can calculate the density loss function $\mathcal{L}_{\mathrm{den}}$ in a fully differentiable manner:

$$\mathcal{L}_{\mathrm{den}} = 1 - \frac{\hat{\boldsymbol{\rho}}(\vec{\boldsymbol{m}}, \vec{\mathbf{x}}) \cdot \boldsymbol{\rho}(\vec{\boldsymbol{m}})}{||\hat{\boldsymbol{\rho}}(\vec{\boldsymbol{m}}, \vec{\mathbf{x}})|| \cdot ||\boldsymbol{\rho}(\vec{\boldsymbol{m}})||} \tag{5}$$

in which the second term is effectively the cosine similarity between $\hat{\boldsymbol{\rho}}(\vec{\boldsymbol{m}}, \vec{\mathbf{x}})$ and $\boldsymbol{\rho}(\vec{\boldsymbol{m}})$. This Gaussian scattering formula to generate the synthetic density map conceptually follows the *molmap* method in UCSF ChimeraX(Pettersen et al., 2021), but the key difference is that we redesign the entire computation process to be fully differentiable using *PyTorch*. This fully differentiable implementation is essential for our method because it allows the density loss $\mathcal{L}_{\mathrm{den}}$ to serve as a restraint term for back-propagation during each refinement loop. It's also worth pointing out that our density generator achieves better performance than the UCSF ChimeraX(Pettersen et al., 2021) it molmap method, with the averaged correlation coefficients between synthetic and input density maps improved from 0.803 to 0.892.

### 3.2.2 GEOMETRY LOSS

Geometry loss functions are employed to guarantee the stereochemical accuracy and structural validity of predicted proteins by enforcing conformity with established biological restraints. However, previous work has failed to incorporate geometry-related losses such as those for Ramachandran plot, rotamer, and $C_\beta$ deviation. Therefore, we innovatively implemented these differentiable geometry loss functions (ramachandran loss $\mathcal{L}_{\mathrm{rama}}$, rotamer loss $\mathcal{L}_{\mathrm{rot}}$, $C_\beta$ deviation loss) in this work.

**Ramachandran loss** intends to exert Ramachandran plot restraints on the predicted protein structures. When computing the Ramachandran loss, all the backbone dihedral angles $\phi$ and $\psi$ will be calculated and evaluated against the Ramachandran criteria retrieved from the Top8000 dataset used by MolProbity(Hintze et al., 2016):

$$\mathcal{L}_{\mathrm{rama}} = \sum_{i=1}^{N_{\mathrm{res}}} \mathbf{F}_{\phi,\psi}(\vec{\boldsymbol{x}}_{i-1}, \vec{\boldsymbol{x}}_i, \vec{\boldsymbol{x}}_{i+1}, a_{i-1}, a_i, a_{i+1}), \tag{6}$$

where $a_i$ denotes the $i^{\mathrm{th}}$ residue's amino acid type and $\mathbf{F}_{\phi,\psi}$ is an indicator function (Appendix E.1) that evaluates whether the backbone tripeptide dihedral angles $\phi_i$ and $\psi_i$ will fall into the outlier region of the Ramachandran plot.

**Rotamer loss** intends to introduce side-chain rotamer-specific restraints:

$$\mathcal{L}_{\mathrm{rot}} = \sum_{i=1}^{N_{\mathrm{res}}} \mathbf{F}_{\chi}(\chi_i, a_i), \tag{7}$$

where $a_i$ denotes the $i^{\mathrm{th}}$ residue's amino acid type and $\mathbf{F}_{\chi}$ is an indicator function (Appendix E.1) that evaluates whether the side-chain structure of the $i^{\mathrm{th}}$ residue (determined by the torsion angles $\chi_i \in (-\pi, \pi]^4$) is an outlier compared to idealized protein secondary-structure fragments from the Top8000 dataset.

The necessity to introduce $C_\beta$ **deviation loss** stems from the importance of $C_\beta$ atoms, which are side-chain carbon atoms bonded to $C_\alpha$ atoms in each amino acid (except glycine). $C_\beta$ atoms are crucial for describing side-chain orientations, indicating potential incompatibility between protein backbones and side-chains. When the actual position of $C_\beta$ atom diverges from the ideal position by over $0.25\mathring{A}$, it is thus considered as a deviation:

$$\mathcal{L}_{\mathrm{C}_\beta} = \sum_{i=1}^{N_{\mathrm{res}}} \mathbf{1}(|\vec{\boldsymbol{x}}_{c_\beta} - \vec{\boldsymbol{x}}'_{c_\beta}| > 0.25), \tag{8}$$

where $\boldsymbol{x}_{c_\beta} \in \mathbb{R}^3$ is the coordinates of predicted $C_\beta$ atoms while $\boldsymbol{x}'_{c_\beta} \in \mathbb{R}^3$ is the idealized $C_\beta$ position computed from positions of other 3 atoms on the same amino acid, i.e. $C_\alpha$, backbone $C$ and backbone $N$.

## 4 EXPERIMENTS

### 4.1 BENCHMARK

We curated a benchmark of 120 complexes (110 protein, 10 DNA/RNA–protein) with cryo-EM density map resolutions ranging from 1.8 Åto 5.9 Å. For each case, initial atomic models were predicted by *AlphaFold3* and subsequently refined against the experimental density map by *Phenix.real_space_refine* and *CryoNet.Refine*. Dataset statistics are provided in Figure 10 and Appendix B.

### 4.2 REFINEMENT OF PROTEIN COMPLEXES

We benchmarked *CryoNet.Refine* against the conventional *Phenix.real_space_refine*. As summarized in Table 1, *CryoNet.Refine* consistently achieves superior scores in both model–map correlation coefficients (CC) and model geometric metrics.

Table 1: Performance on benchmark of 110 protein complexes(↑: higher is better, ↓: lower is better).

| Category | Metrics | *AlphaFold3* | *Phenix.real_space_refine* | *CryoNet.Refine* |
|---|---|---|---|---|
| Model-map Correlation Coefficients | $CC_{mask}$ ↑ | 0.38 | 0.54 | **0.59** |
| | $CC_{box}$ ↑ | 0.41 | 0.53 | **0.57** |
| | $CC_{mc}$ ↑ | 0.40 | 0.55 | **0.60** |
| | $CC_{sc}$ ↑ | 0.39 | 0.55 | **0.58** |
| | $CC_{peaks}$ ↑ | 0.27 | 0.40 | **0.45** |
| | $CC_{volume}$ ↑ | 0.42 | 0.55 | **0.60** |
| Model Geometric Metrics | Angle RMSD (degree)↓ | 1.58 | 0.72 | **0.36** |
| | $C_\beta$ deviations↓ | 0.03 | **0.00** | 0.00 |
| | Ramachandran favored (%)↑ | 95.73 | 96.39 | **98.92** |
| | Ramachandran outlier (%)↓ | 0.82 | **0.02** | 0.06 |
| | Rotamer favored (%)↑ | 97.08 | 85.42 | **98.64** |
| | Rotamer outlier (%)↓ | 1.08 | 1.15 | **0.49** |

Figure 3 and Figure 4 show the performance of *CryoNet.Refine* across six model-map correlation co-efficients (CC) and model geometric metrics, respectively. Notably, the consistent gains in $CC_{mask}$ (0.59 vs. 0.54) and $CC_{mc}$ (0.60 vs. 0.55) suggest the refined atomic model achieves more accurate main-chain placement within the density map. Regarding geometry, our method consistently achieves superior stereochemistry, virtually eliminating $C_\beta$ deviations, raising Ramachandran favored percentage to nearly 99% (98.92%), and reducing rotamer outliers by approximately 57% compared to *Phenix.real_space_refine* (1.15% to 0.49%). Furthermore, Angle RMSD is drastically reduced from the initial 1.58° to 0.37°. These results emphatically highlight that *CryoNet.Refine* simultaneously improves the stereochemical geometry and the agreement to the cryo-EM density map.

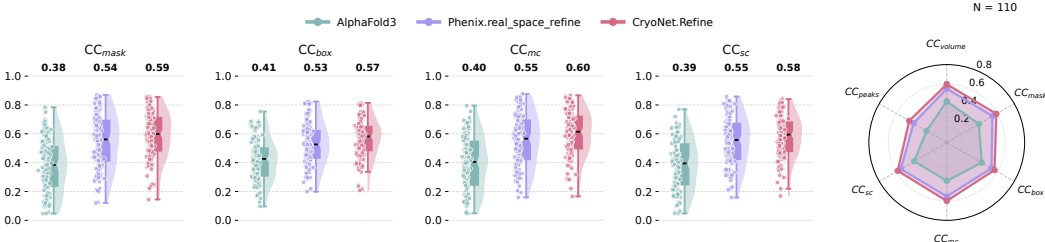

Figure 3: Model–map correlation coefficients on protein complex benchmark.

Figure 5 shows that *CryoNet.Refine* outperforms both *AlphaFold3* and *Phenix.real_space_refine* in model-map correlation coefficients and model geometric metrics.

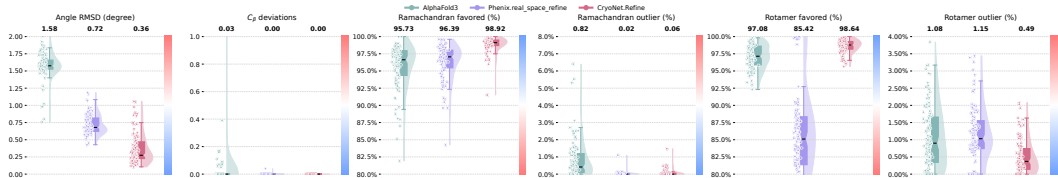

Figure 4: Model geometric metrics. (Color gradient: blue for better, red for worse.)

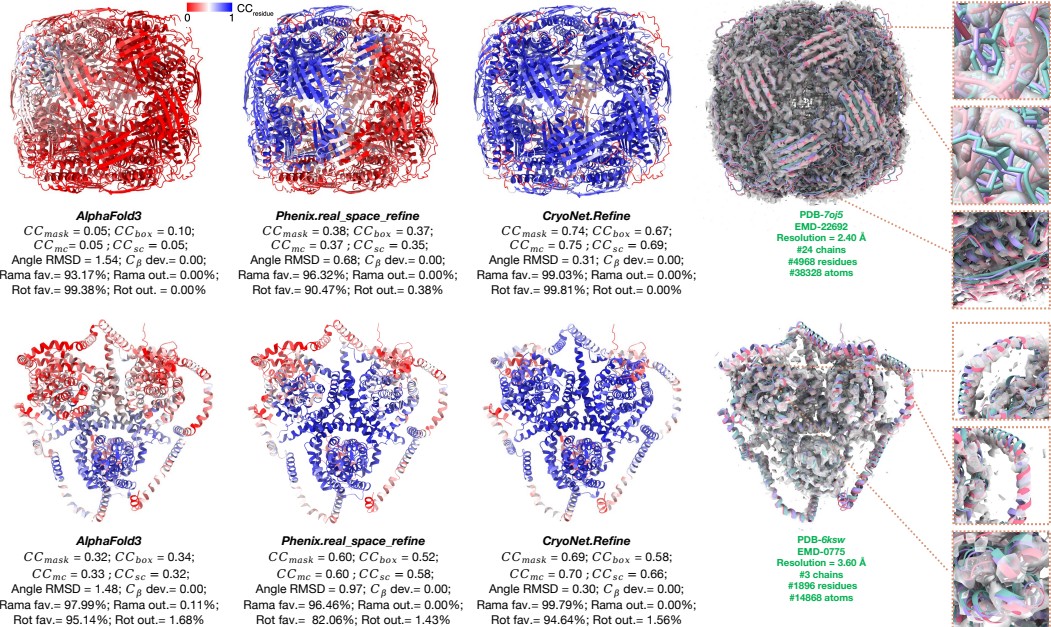

Figure 5: The input atomic models from *AlphaFold3*, the refined atomic model from *Phenix. real_space_refine* and *CryoNet.Refine* on the Medicago truncatula HISN5 protein (PDB-*7oj5*; EMD-22692) and the human concentrative nucleoside transporter CNT3 (PDB-*6ksw*, EMD-0775) complex. Inserts in the right panel show that the main-chains and side-chains generated from *CryoNet. Refine* model align well with the density map.

## 4.3 REFINEMENT ON DNA/RNA–PROTEIN COMPLEXES

We next evaluated *CryoNet.Refine* on DNA/RNA–protein complex. As our current implementation doesn't incorporate nucleic acid–specific stereochemical restraints, the assessment exclusively focused on model–map correlation coefficients (CC) as shown in Table 2. Across all metrics, *CryoNet.Refine* consistently outperforms both *AlphaFold3* and *Phenix.real_space_refine*, demonstrating substantially improved agreement with experimental densities.

Table 2: Performance on DNA/RNA–protein complexes.

| Category | Metrics | *AlphaFold3* | *Phenix.real_space_refine* | *CryoNet.Refine* |
|---|---|---|---|---|
| Model–map Correlation Coefficients | $CC_{mask}$ ↑ | 0.40 | 0.57 | **0.65** |
| | $CC_{box}$ ↑ | 0.49 | 0.61 | **0.67** |
| | $CC_{mc}$ ↑ | 0.45 | **0.62** | 0.61 |
| | $CC_{sc}$ ↑ | 0.42 | 0.58 | **0.67** |
| | $CC_{peaks}$ ↑ | 0.35 | 0.51 | **0.60** |
| | $CC_{volume}$ ↑ | 0.48 | 0.61 | **0.69** |

For DNA/RNA–protein complex, we show model–map correlation coefficients in Figure 6. It can be observed that *CryoNet.Refine* achieved substantial improvements in both $CC_{mask}$ and $CC_{sc}$ , strongly indicating superior performance refinement in both main-chain and side-chain. A representative case study is shown in Figure 7, where *AlphaFold3* and *Phenix.real_space_refine* achieve moderate density fitting ($CC_{mask}$ = 0.18 and 0.36, respectively), whereas *CryoNet.Refine* attains a markedly higher $CC_{mask}$ = 0.72, along with consistent gains across all other CC metrics.

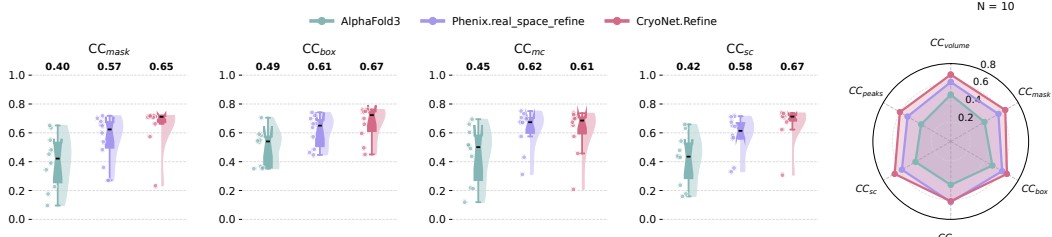

Figure 6: Model–map correlation coefficients on DNA/RNA-protein complex.

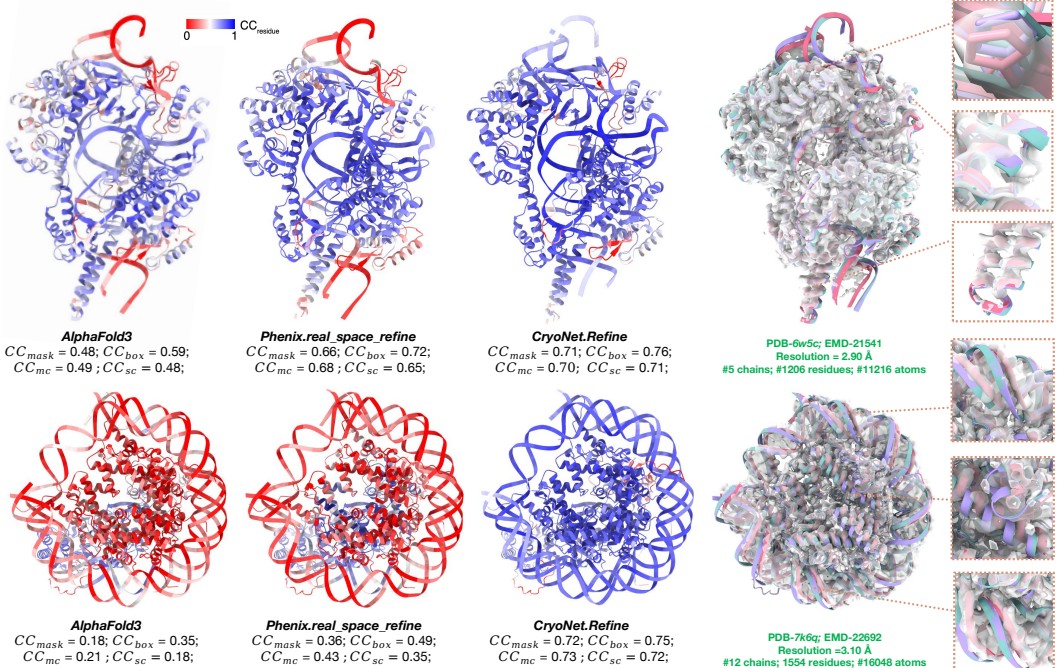

Figure 7: The input atomic models from *AlphaFold3*, the refined atomic models from *Phenix. real_space_refine* and *CryoNet.Refine* are shown on the Cas12i(E894A)–crRNA–dsDNA (PDB-*6w5c*, EMD-21541) and the active-state Dot1 bound to the H4K16ac nucleosome (PDB-*7k6q*, EMD-22692) complex. Inserts in the right panel show that the main-chains and side-chains generated by *CryoNet.Refine* align well with the density map.

## 5 ABLATION STUDY

Results of the ablation study can be found in Appendix F.3. The full *CryoNet.Refine* configuration achieves the best balance between model–map correlation coefficients and model geometric metrics. Removing individual loss terms leads to distinct degradations in correlation metrics or geometric metrics, confirming the complementary roles of density and geometry loss.

### 5.1 RUNTIME PERFORMANCE

We benchmarked the runtime of *CryoNet.Refine* against *Phenix.real_space_refine*. *CryoNet.Refine* consistently achieves highly efficient performance, whereas *Phenix.real_space_refine*'s CPU-only support incurs higher computational costs, particularly for large complexes.

Across 120 complexes, *CryoNet.Refine* ran faster than *Phenix.real_space_refine* in 65 cases (54.2%). These results underscore that our method combines superior accuracy with high efficiency, making it ideal for large-scale, high-throughput cryo-EM refinement.

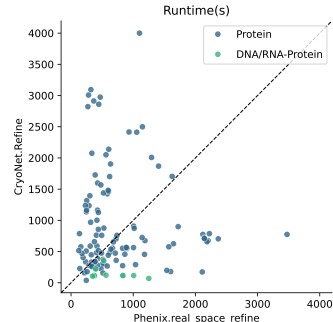

Figure 8: Runtime comparison.

## 6  CONCLUSION

In this work, we present *CryoNet.Refine*, a pioneering one-step diffusion module for cryo-EM atomic model refinement. It critically features a novel differentiable density loss and a comprehensive geometry metric loss, achieving unparalleled refinement accuracy.

Our framework opens new avenues for the field: the density loss can be broadly applied to AI-based atomic model building, while the geometry loss offers essential guidance for protein structure prediction. Specifically, these differentiable constraints are model-agnostic and can complement the implicit priors of foundation models like *AlphaFold3*. By integrating them into conditional diffusion sampling strategies, future research could bridge the gap between generative priors and the atomic-level stereochemical precision required for high-resolution structures.

*CryoNet.Refine* thus marks a significant advance for the cryo-EM community. Looking ahead, we aim to extend the specialized geometry loss for DNA/RNA structures and incorporate a steric clash loss. We also acknowledge the computational cost associated with per-structure optimization; therefore, future work will focus on enhancing efficiency through parallel refinement frameworks and faster convergence strategies. Ultimately, we intend to demonstrate superior performance on challenging low-resolution density maps, establishing *CryoNet.Refine* as an essential tool for structural biology.

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

## A  NOTATION

### Framework

| | |
|---|---|
| $d_0$ | Input cryo-EM density map |
| $x_0$ | All-atom coordinates of input structure |
| $s$ | Sequence Embedding |
| $z$ | Pairwise Representation |
| $x_i(i = 1, \cdots, n)$ | Refined atomic model after the $i$ recycle |
| $d_i(i = 1, \cdots, n)$ | Synthetic density map of refined atomic model $x_i$ |
| $\gamma_t$ | Weight for a loss function term $\mathcal{L}_t$ |

### Diffusion Module

| | |
|---|---|
| $\xi_0, \xi_{min}$ | Noise scaling hyperparameter |
| $\sigma$ | Noise scale parameter in diffusion refinement |
| $\sigma_{\text{data}}$ | Data-dependent scale constant (fixed to 16), estimated from training distribution |
| $c_{\text{skip}}(\sigma)$ | Skip coefficient in preconditioning |
| $c_{\text{out}}(\sigma)$ | Output scaling coefficient in preconditioning |
| $c_{\text{in}}(\sigma)$ | Input scaling coefficient in preconditioning |
| $c_{\text{noise}}(\sigma)$ | Noise embedding term in preconditioning, |
| $\mathcal{C}$ | Auxiliary conditioning features derived from $s$, $z$, and encoded structural features |

### Density Loss

| | |
|---|---|
| $w$ | Atomic weight |
| $res$ | Resolution of the input density map |
| $v$ | Voxel size of the input density map |
| $r_{\mathcal{G}}$ | Gaussian radius |
| $s_{\mathcal{G}}$ | Radius of the virtual Gaussian sphere |
| $L$ | Number of grid points for each atom per axis |
| $\vec{\mathbf{x}} = [\mathbf{x}^0, \mathbf{x}^1, \mathbf{x}^2]$ | All-atom coordinates of refined atomic models |
| $d = [\vec{\boldsymbol{m}}, \boldsymbol{\rho}]$ | Synthetic density map |
| $\vec{\boldsymbol{m}} = [\boldsymbol{\lambda}, \boldsymbol{\mu}, \boldsymbol{\nu}]$ | Grid points of density maps |
| $\hat{\boldsymbol{\rho}}$ | Synthetic density values |
| $\boldsymbol{\rho}$ | Density values retrieved from the input density map |

### Geometry Loss

| | |
|---|---|
| $a$ | Amino acid type |
| $N_{\text{res}}, N_{\text{atom}}, N_{\text{bond}}$ | Number of residues, atoms and bonds in a structure |
| $\theta$ | Bond angles |
| $\chi$ | Torsion angles |

## B  DATASET

Figure 10 summarizes the dataset statistics. The two panels respectively show the number of chains and the number of residues with respect to map resolution. Protein-only complexes display broader diversity in both chain counts and residue numbers, whereas DNA/RNA–protein assemblies tend to be smaller but fall within comparable resolution ranges.

For all targets, input sequences were retrieved from the RCSB Protein Data Bank (PDB). Since *AlphaFold3* imposes a practical sequence length limit of $\leq 5000$ amino acids, we adopted a chain-wise strategy: sequences were segmented at the chain level and truncated if necessary, ensuring that the combined length per case did not exceed 5000 residues. This preprocessing step guarantees compatibility with *AlphaFold3* while preserving the completeness of structural contexts across chains.

For residues with backbone correlation coefficients below 0.1, we applied the ADP-EM local refinement tool (Garzon et al., 2007) to pre-process the models and improve rigid-body docking.

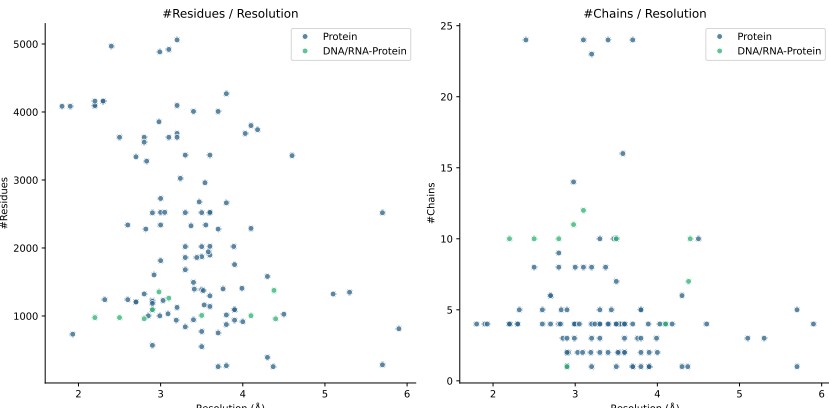

Figure 9: Dataset statistics of density map resolution, number of chains, and number of residues.

## C  ONE-STEP DIFFUSION MODULE

Classical diffusion models define a forward noising process and a multi-step reverse denoising chain (Ho et al., 2020; Song et al., 2020). While powerful, such iterative schemes incur substantial computational cost. Inspired by accelerated sampling approaches such as consistency models (Song et al., 2023) and diffusion distillation (Meng et al., 2022), we reformulated diffusion into a *deterministic one-step refinement* tailored for cryo-EM structural optimization.

Unlike conventional diffusion that begins from Gaussian noise, we initialize directly from the the starting structure $\mathbf{x}_0$.

Refinement is then performed through a **preconditioned forward module**(Karras et al., 2022):

$$\mathbf{x}_i = c_{\text{skip}}(\sigma)\,\mathbf{x}_0 \;+\; c_{\text{out}}(\sigma)\,\mathcal{F}_\theta\big(c_{\text{in}}(\sigma)\mathbf{x}_0,\, c_{\text{noise}}(\sigma),\, \mathcal{C}\big), \tag{9}$$

with auxiliary conditioning $\mathcal{C}$. $\mathcal{F}_\theta$ is a neural network responsible for the scoring mechanism. The other coefficients include:

$$c_{\text{skip}}(\sigma) = \frac{\sigma_{\text{data}}^2}{\sigma^2 + \sigma_{\text{data}}^2}, \qquad\qquad c_{\text{out}}(\sigma) = \frac{\sigma\,\sigma_{\text{data}}}{\sqrt{\sigma^2 + \sigma_{\text{data}}^2}}, \tag{10}$$

$$c_{\text{in}}(\sigma) = \frac{1}{\sqrt{\sigma^2 + \sigma_{\text{data}}^2}}, \qquad\qquad c_{\text{noise}}(\sigma) = \tfrac{1}{4}\log\Big(\tfrac{\sigma}{\sigma_{\text{data}}}\Big). \tag{11}$$

The refined structure after the $i^{th}$ recycle is obtained in a single deterministic step:

$$\mathbf{x}_i = \Phi_\theta(\mathbf{x}_0, \sigma, \mathcal{C}) \tag{12}$$

where the deterministic refinement operator $\Phi_\theta$ is instantiated using the preconditioned forward update (Eq. 9):

$$\Phi_\theta(\mathbf{x}_i, \sigma, \mathcal{C}) = c_{\text{skip}}(\sigma)\, \mathbf{x}_0 + c_{\text{out}}(\sigma)\, \mathcal{F}_\theta\big(c_{\text{in}}(\sigma)\mathbf{x}_0,\, c_{\text{noise}}(\sigma),\, \mathcal{C}\big). \tag{13}$$

---

**Algorithm 1** One-step Deterministic Refinement in *CryoNet.Refine*

---

**Require:** Initial structure $\mathbf{x}_0$, conditioning $\mathcal{C}$
**Require:** Hyperparameters: $\xi_0$, $\xi_{\min}$, $\sigma$, learning rate $\eta$
**Require:** Number of recycling steps $n$
**Ensure:** Refined structure $\mathbf{x}_{\text{final}}$
 1: Initialize model parameters $\theta$
 2: **for recycle** $i = 1, 2, \ldots, n$ **do**
 3:   **Step 1: Compute noise level adjustment**
 4:   $\xi \leftarrow (\sigma > \xi_{\min})\, ?\, \xi_0 : 0$
 5:   $\hat{t} \leftarrow \sigma(1 + \xi)$
 6:   **Step 2: Perform structure update (One-step)**
 7:   $\mathbf{x}_{\text{pred}} \leftarrow \Phi_\theta(\mathbf{x}_0, \hat{t}, \mathcal{C})$
 8:   **Step 3: Compute loss and backpropagate**
 9:   $\mathcal{L} \leftarrow \mathcal{L}_{\text{density}}(\mathbf{x}_{\text{pred}}, \mathcal{C}) + \mathcal{L}_{\text{geometry}}(\mathbf{x}_{\text{pred}})$
10:   $\theta \leftarrow \theta - \eta \nabla_\theta \mathcal{L}$
11:   **if** converged **then** break
12: **end for**
13: **return** $\mathbf{x}_{\text{pred}}$

---

Compared with *AlphaFold3*, which employs multi-step stochastic denoising from Gaussian initialization (Abramson et al., 2024), our formulation collapses the diffusion chain into a single deterministic refinement. This yields (i) efficiency by eliminating iterative sampling, (ii) stability via input–output preconditioning, and (iii) broad applicability to proteins, DNA/RNAs and complexes.

It is important to distinguish between the diffusion sampling mechanism and the workflow. The term **"One-step"** refers to the efficiency of our diffusion sampler (Eq. 13). Unlike stochastic samplers that require hundreds of iterative denoising steps to generate a structure, our preconditioned module maps the initial structure $\mathbf{x}_0$ directly to the predicted refined state in a single forward pass.

In contrast, **"Recycling"** refers to the outer test-time optimization loop. Although the sampler is capable of generating a structure in one step, achieving high-precision fit to experimental density requires iterative tuning. In each recycle, we perform a one-step forward pass using the fixed input $\mathbf{x}_0$, compute density and geometry losses on the output, and backpropagate gradients to **update the model parameters** $\theta$. This iterative optimization allows the diffusion module to progressively customize its weights to the specific instance, minimizing the loss over multiple cycles.

## C.1 ARCHITECTURAL DIFFERENCES FROM ALPHAFOLD3

While *CryoNet.Refine* initializes its parameters from the weights of *Boltz-2*, its architectural design and inference workflow diverge fundamentally from the standard AlphaFold3 framework to address the specific challenges of cryo-EM refinement. The key distinctions are summarized below:

1. **Input Initialization:** Standard *AlphaFold3* is a generative model designed for *de novo* prediction, initializing the diffusion process from Gaussian noise ($\mathbf{x}_T \sim \mathcal{N}(\mathbf{0}, \mathbf{I})$) to generate diverse conformations. In contrast, *CryoNet.Refine* is a refinement framework. We initialize directly from the initial atomic model, treating it as a noisy state that requires correction against the density map.

2. **Sampling Trajectory:** AlphaFold3 employs a stochastic multi-step sampler to traverse the reverse diffusion process, which is computationally expensive and introduces randomness. *CryoNet.Refine* reformulates this into a one-step prediction, mapping the input state directly to the denoised structure in a single pass for efficiency.

3. **Optimization Paradigm:** AlphaFold3 typically operates in a fixed-weight inference mode. Conversely, *CryoNet.Refine* adopts a test-time optimization strategy. During the refinement loop (Algorithm 1), we explicitly compute gradients from the density and geometry losses

to update the diffusion module parameters ($\theta$). This allows the network to learn the specific features of the experimental density map for each individual case.

4. **Module Adaptations and Trainability:** To specialize for density-guided refinement, we streamlined the modular architecture.

- Multi-sequence alignment (MSA) processing, confidence heads (pLDDT/PAE), and internal architecture-level recycling are removed, as the method relies on physical density constraints rather than evolutionary information.
- The **Sequence Embedder, Atom Encoder, and Pairformer** are kept **frozen** to preserve robust structural priors derived from pre-training. Only the **Diffusion Module** is trainable, focusing the optimization capacity solely on coordinate adjustment driven by the experimental map.

## D  DENSITY GENERATOR

Our current density generator builds the target density volume by summing Gaussian functions centered on atomic positions with the simulation process specifically tailored to the target resolution. This procedure conceptually follows the *molmap* implementation in UCSF ChimeraX(Pettersen et al., 2021).

A density map $d$ can be discretized as a group of 4-dimensional vectors $[\boldsymbol{\lambda}, \boldsymbol{\mu}, \boldsymbol{\nu}, \boldsymbol{\rho}]$, where $\vec{\boldsymbol{m}} = [\boldsymbol{\lambda}, \boldsymbol{\mu}, \boldsymbol{\nu}]$ act as grid points that can specify a certain location and $\boldsymbol{\rho}$ is the density values at each grid point, with the origin of coordinates nearest to $\vec{\boldsymbol{m}}_{\min}$ and farthest to $\vec{\boldsymbol{m}}_{\max}$. For computation of the overlapped region, we first locate this region $\forall \vec{\boldsymbol{m}}_o \in \mathcal{S}_o$ between the input density map $d_0$ and the synthetic density map $d_i$ through the coordinate system:

$$\vec{\boldsymbol{m}}_{o_{\min}} = \max(\vec{\boldsymbol{m}}_{0_{\min}}, \vec{\boldsymbol{m}}_{i_{\min}}) \quad \vec{\boldsymbol{m}}_{o_{\max}} = \min(\vec{\boldsymbol{m}}_{0_{\max}}, \vec{\boldsymbol{m}}_{i_{\max}})$$

After locating $\mathcal{S}_o$, we can select all the density values within it from $d_0$ and $d_i$ and get their subsets $d_{0 \cap o}$ and $d_{i \cap o}$. Thus, the in this overlapped region $\mathcal{S}_o$, our synthetic density values and the input density values will be defined as:

$$\hat{\boldsymbol{\rho}} = \boldsymbol{\rho}_{0 \cap o}, \quad \boldsymbol{\rho} = \boldsymbol{\rho}_{i \cap o}$$

For the implementation of synthetic density map generation, we follow the *molmap* algorithm in ChimeraX(Pettersen et al., 2021), utilizing the all-atom coordinates of the refined atomic model at the $i^{th}$ recycle $\vec{\boldsymbol{x}}_i = [\boldsymbol{x}_i^0, \boldsymbol{x}_i^1, \boldsymbol{x}_i^2]$ and output a box-shaped density map $d_i = (\vec{\boldsymbol{m}}_i, \boldsymbol{\rho}_i)$, where $\vec{\boldsymbol{x}}_i \in \mathbb{R}^{3 \times N_{\text{atom}}}$ and $\vec{\boldsymbol{m}}_i \in \mathbb{R}^{3 \times N_{\text{atom}} \times L}$, noting that the parameter $L$ stands for the number of grid points for each atom per axis, and its value is obtained by:

$$r_{\mathcal{G}} = res/(\pi \cdot v), \quad s_{\mathcal{G}} = 4 \cdot r_{\mathcal{G}}, \quad L = 2 \cdot s_{\mathcal{G}}$$

where $res$ is the input density map resolution and $v$ represents its voxel size. The key idea here is to construct a Gaussian sphere for each atom, with $r_{\mathcal{G}}$ and $s_{\mathcal{G}}$ respectively meaning Gaussian radius and the radius of a Gaussian sphere. We expect that $\vec{\boldsymbol{m}}_i$ turns out to be a box as small as possible with all-atom coordinates $\vec{\boldsymbol{x}}_i$ inside it as well as a margin of Gaussian sphere around it, so:

$$\vec{\boldsymbol{m}}_{i_{\min}} = [\lfloor \min(\boldsymbol{x}_i^0) \rfloor, \lfloor \min(\boldsymbol{x}_i^1) \rfloor, \lfloor \min(\boldsymbol{x}_i^2) \rfloor], \quad \vec{\boldsymbol{m}}_{i_{\max}} = [\lceil \max(\boldsymbol{x}_i^0) \rceil, \lceil \max(\boldsymbol{x}_i^1) \rceil, \lceil \max(\boldsymbol{x}_i^2) \rceil]$$
$$\vec{\boldsymbol{m}}_{i\_\text{size}} = \lceil (\vec{\boldsymbol{m}}_{i_{\max}} - \vec{\boldsymbol{m}}_{i_{\min}})/v + 2 \cdot s_{\mathcal{G}} + 1 \rceil, \quad \vec{\boldsymbol{m}}_{i_{\text{center}}} = (\vec{\boldsymbol{m}}_{i_{\min}} + \vec{\boldsymbol{m}}_{i_{\max}})/(2 \cdot v)$$
$$\vec{\boldsymbol{m}}_{i_{\min}} = \lfloor (\vec{\boldsymbol{m}}_{i_{\text{center}}} - \vec{\boldsymbol{m}}_{i\_\text{size}})/2 \rfloor, \quad \vec{\boldsymbol{m}}_{i_{\max}} = \lfloor (\vec{\boldsymbol{m}}_{i_{\text{center}}} + \vec{\boldsymbol{m}}_{i\_\text{size}})/2 \rfloor$$

For each atom per axis(for example the x-axis), we can get $L$ density values around the original atom coordinate by applying Gaussian distribution:

$$\text{for } n \in (1, 2, ..., N_{\text{atom}}) \quad \boldsymbol{\rho}_\lambda^n = w_n \cdot \exp(-\frac{(\boldsymbol{x}_i^0)^n + \boldsymbol{l}}{2}), \boldsymbol{l} = [-s_{\mathcal{G}}, ..., 0, -s_{\mathcal{G}}], \tag{14}$$

where $w_n$ signifies the atomic weight of the corresponding atom. To combine density values on all directions, each atom will end up with $L^3$ density values:

$$\vec{\boldsymbol{\rho}}^n = \boldsymbol{\rho}_\lambda^n \times \boldsymbol{\rho}_\mu^{n\top} \times \boldsymbol{\rho}_\nu^{n\top} \tag{15}$$

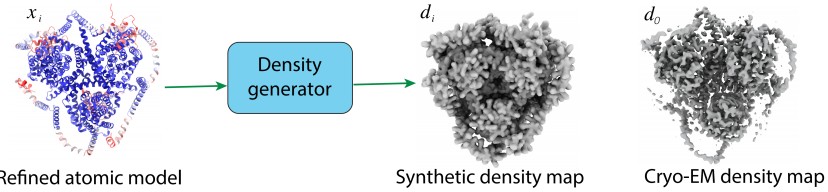

Figure 10: Data Patterns and Visualizations of the Density Generator Output

The key difference between *molmap* and our density generator is that we redesign the entire computation to be fully differentiable using *PyTorch*, which is essential for our method because it allows the density loss (controlled by a single hyperparameter, $\gamma_{den}$) to serve as a restraint term for back-propagation during the refinement loop.

To evaluate how well the simulated maps agree with input experimental maps, we calculate their correlation coefficients across all cases. The average correlation is 0.892, significantly improving from 0.803 when using the USCF ChimeraX *molmap* method. Nevertheless, simulated maps inevitably fall short of capturing the complex characteristics introduced by diverse experimental instruments and imaging conditions. In particular, secondary structure elements, map artifacts, and noise patterns in our simulated densities differ from those observed in experimental maps. To address these limitations, future work will explore replacing the simulation-based density generator with a deep learning–based generative model, which could better reproduce the statistical and structural properties of experimental cryo-EM maps and yield a more informative and accurate density loss.

# E  LOSS FUNCTIONS

After collective parameter tuning, the weight values for each component of our loss function are listed in Table 3 .

Table 3: Loss function weights

| Loss Functions | Weight | Value |
|---|---|---|
| Density loss ($\mathcal{L}_{\text{den}}$) | $\gamma_{\text{den}}$ | 20.0 |
| Ramachandran loss ($\mathcal{L}_{\text{rama}}$) | $\gamma_{\text{rama}}$ | 500 |
| Rotation loss ($\mathcal{L}_{\text{rot}}$) | $\gamma_{\text{rot}}$ | 500 |
| Angle loss ($\mathcal{L}_{\text{angle}}$) | $\gamma_{\text{angle}}$ | 2 |
| $C_\beta$ loss ($\mathcal{L}_{C_\beta}$) | $\gamma_{C_\beta}$ | 50 |
| Violation loss ($\mathcal{L}_{\text{viol}}$) | $\gamma_{\text{viol}}$ | 1000 |

## E.1  RAMACHANDRAN AND ROTAMER LOSS

The Ramachandran plot (Goodman et al., 1970) is a 2D graph with $\phi$ angles and $\psi$ angles as x- and y-axis. By plotting the combination of these torsion angles of all residues, it provides insights into whether the conformation is sterically favored, allowed or outliered.

In order to obtain more updated and more representative protein structures for reference, MolProbity(Hintze et al., 2016), the widely used validation program for protein structures, curated the Top8000 dataset by filtering and deduplicating high-quality protein structures from the PDB Bank. The release of the Top8000 dataset came with their 3-dimensional Ramachandran grids plotting the ideal distribution for different kinds of amino acids,[1].The z-axis values act as density scores measuring the frequency for a certain combination of Ramachandran angles.

Through interpolation of the nearest grid values, we can get the density score for any combination of arbitrary Ramachandran scores and classify them as falling within favored, allowed, or outlier regions, as shown in Algorithm 2

---

[1]In fact, there are 6 distinct types of Ramachandran plots for different amino acids, including *Gly*, *Val/Ile*, *pre-Pro*, *trans-Pro*, *cis-Pro* and *Ala*. How to differentiate them and the rationale behind this categorization are beyond the scope of this paper.

As for $\mathcal{F}_\chi$ of rotamer loss, the procedure is highly similar to Algorithm 2 except that it requires $\chi$ angles instead of $\phi, \psi$ angles, and it performs 4-dimensional interpolation (since each density value corresponds to the combination of four $\chi$ angles) from the rotamer distribution library of Top8000 dataset.

---

**Algorithm 2** Classification of Ramachandran Angle Outliers

---

$\vec{x}_i$: The i-th residue coordinates
$a_i$: The amino acid type of the i-th residue
$N_{\text{res}}$: total number of residues
$\mathcal{S}_{\text{rama\_type}} = \{$*Gly, Val/Ile, pre-Pro, trans-Pro, cis-Pro, Ala*$\}$
$r_i \in \mathcal{S}_{\text{rama\_type}}$: categorization Ramachandran angles between $n_{i-1}, n_i, n_{i+1}$,
$\tau_r$: density value threshold for Ramachandran category $r$
$o$: count of Ramachandran outliers
**Input** $\vec{x} = [\vec{x}_1, \vec{x}_2, \cdots, \vec{x}_{N_{\text{res}}}], [a_1, a_2, \cdots, a_{N_{\text{res}}}]$
**Function** *getDensity*$(\phi, \psi, r_i)$:
$\mathcal{R} \leftarrow$ Ramachandran distribution grids for $r_i$
density\_score $\leftarrow$ 2DInterpolate$(\mathcal{R}(\lfloor\phi\rfloor, \lfloor\psi\rfloor), \mathcal{R}(\lceil\phi\rceil, \lceil\psi\rceil))$
**return** density\_score
**for** $i = 2$ to $N_{\text{res}} - 1$
$\phi_i, \psi_i \leftarrow$ *calculateDihedrals*$(\vec{x}_{i-1}, \vec{x}_i, \vec{x}_{i+1})$
$r_i \leftarrow$ RamaType$(a_{i-1}, a_i, a_{i+1})$
density\_score$_i \leftarrow$ getDensity$(\phi_i, \psi_i, r_i)$
**if** density\_score$_i < \tau_r$ **then**
    $o \leftarrow o + 1$
**end if**
**return** $o/N_{\text{res}}$

---

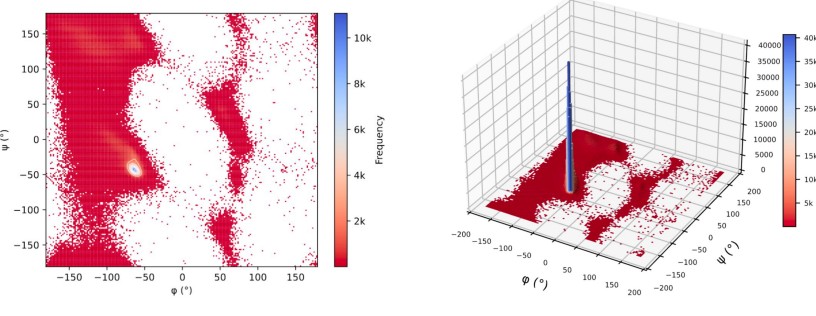

(a) 2D Ramachandran plot    (b) 3D Ramachandran distribution

Figure 11: Visualization of Ramachandran distributions

### E.2 BOND ANGLE LOSS

**Bond angle loss** applies geometry restraints on chemical bonds and forces them to approach the target values specified by the ideal geometry library. In our implementation, $\mathcal{L}_{\text{angle}}$ is essentially equivalent to the structure's bond angle RMSD value:

$$\mathcal{L}_{\text{anlge}} = \sqrt{\frac{1}{N_{\text{bond}}} \sum_{i=1}^{N_{\text{bond}}} \Delta\theta_i^2}, \tag{16}$$

where $N_{\text{bond}}$ counts the number of all chemical bonds inside the whole protein structure, and $\Delta\theta_i \in (-\pi, \pi]$ calculates the minimal angular difference between the predicted angle value $\theta_i$ and the ideal angle value $\theta_i'$:

$$\Delta\theta_i = \arctan(\sin(\theta_i - \theta_i')/\cos(\theta_i - \theta_i')) \tag{17}$$

To identify chemical bonds given all-atom coordinates, we follow the algorithm as well as ideal bond angle values given by *phenix.pdb_interpretation*(Afonine et al., 2012) and rewrite the implementation using PyTorch.

### E.3 VIOLATION LOSS

**Violation loss** penalizes steric clashes between nonbonded atoms to prevent potential clashes. We borrow the OpenFold implementation (Ahdritz et al., 2024) and follow the definition of (Jumper et al., 2021):

$$\mathcal{L}_{\text{viol}} = \sum_{i=1}^{N_{\text{nbpairs}}} \max(s_{lit}^i - \tau - s_{pred}^i, 0), \tag{18}$$

where $s_{pred}^i$ is the distance of two non-bonded atoms in the predicted structure and $s_{lit}^i$ is the "clashing distance" of these two atoms according to their literature Van der Waals radii. $N_{\text{nbpairs}}$ is the number of all nonbonded atom pairs in this structure. The tolerance $\tau$ is set to 1.5 Å.

## F RESULTS

### F.1 RECYCLE STRATEGY

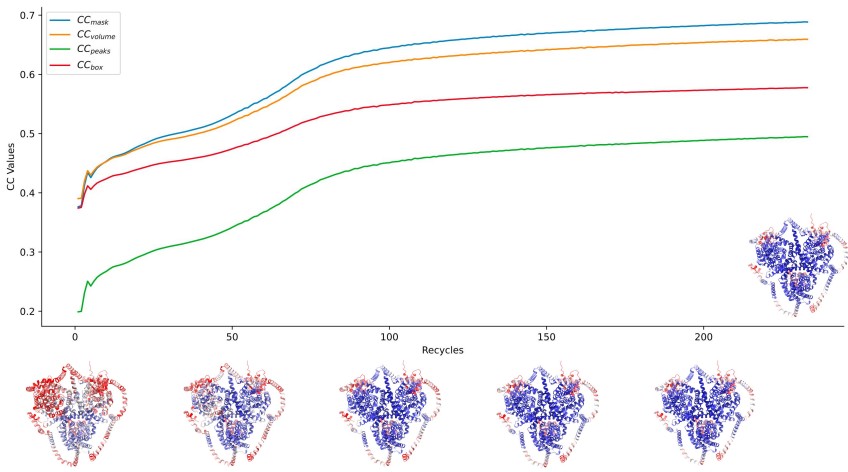

Figure 12: Model-map Correlation Coefficients (CC) analysis of protein complexes across different recycling numbers (We provide a trajectory video over the entire 234 refinement recyclings in supplementary material).

The recycling strategy of *CryoNet.Refine* sets a maximum of 300 iterations combined with an early-stop mechanism. To demonstrate the rationale behind this, we picked a protein structure from our dataset (PDB-*6ksw*) and trace its model-map correlation coefficients (CC) values after each recycle iteration, as shown in Figure 12.

This trajectory serves as a sensitivity analysis regarding the number of recycling steps, revealing two distinct phases in the refinement process:

1. High-Sensitivity Phase (Recycles $0-100$): The CC metrics increase sharply, indicating that the model is highly responsive to the initial density-aware gradient updates as it resolves major structural conflicts.

2. Robust Convergence Phase (Recycles $> 100$): The curves plateau asymptotically, showing that the refinement becomes robust and insensitive to additional iterations once the structure matches the density map constraints.

These 4 CC values basically follow a similar pattern. Consequently, the setting of 300 iterations provides a sufficient safety margin for convergence, while the early-stop mechanism effectively identifies the transition to the plateau phase, avoiding unnecessary computation.

## F.2 DIFFUSION SAMPLING STEPS AND COMPARISON WITH CLASSICAL DIFFUSION

Figure 13: Model-map Correlation Coefficients (CC) analysis of DNA/RNA–protein complex across different diffusion sampling steps.

To show that our one-step diffusion module is an improvement over previous diffusion models with multiple sampling steps, we picked a protein structure (PDB-*6ksw*) and run it with 200 steps. The model-map correlation coefficients (CC) values after each step are shown in Figure 13. Each curve has its optimal value at the beginning step and steadily falls when sampling more steps, which demonstrates the efficacy of our one-step diffusion module.

To validate our design choice of a one-step diffusion module, we conducted a comparative ablation study against a classical multi-step diffusion approach.

**Experimental Setup.** For the classical diffusion baseline, we employed a standard stochastic sampling trajectory with $T = 200$ steps within one recycle, a setting typical for generative protein structure models like *Boltz-2* or *AlphaFold3*. The inference process started from the initial *AlphaFold3* prediction. At each step $t$, the network predicted the denoised structure, and Gaussian noise was injected according to a standard variance schedule before proceeding to step $t - 1$.

**Quantitative Results.** As illustrated in Figure 13 (which shows the rapid decay of CC values over steps) and quantified in Table 4, the multi-step approach leads to a catastrophic degradation in both model-map correlation and geometric integrity. The 200-step sampler yields a $CC_{mask}$ of only 0.30 and introduces severe geometric violations (e.g., Angle RMSD of 1.66°).

Table 4: Quantitative comparison between CryoNet.Refine (One-step) and Classical Diffusion (200 steps) across 27 cases. The multi-step approach fails to preserve structural biology constraints when applied to refinement tasks.

| Category | Metrics | *CryoNet.Refine* (One-step) | Classical Diffusion (200 steps) |
|---|---|---|---|
| Model-map Correlation Coefficients | $CC_{mask}$ ↑ | **0.65** | 0.30 |
| | $CC_{box}$ ↑ | **0.58** | 0.35 |
| | $CC_{mc}$ ↑ | **0.66** | 0.31 |
| | $CC_{sc}$ ↑ | **0.63** | 0.31 |
| | $CC_{peaks}$ ↑ | **0.47** | 0.24 |
| | $CC_{volume}$ ↑ | **0.65** | 0.34 |
| Model Geometric Metrics | Angle RMSD (°)↓ | **0.54** | 1.66 |
| | $C_\beta$ deviations↓ | **0.00** | **0.00** |
| | Ramachandran favored (%)↑ | **98.80** | 98.63 |
| | Ramachandran outlier (%)↓ | **0.10** | 0.81 |
| | Rotamer favored (%)↑ | **98.58** | 98.63 |
| | Rotamer outlier (%)↓ | 0.51 | **0.18** |

We suspect that the degradation arises from 2 main factors: (1) We employ a deterministic sampler, which intrinsically requires fewer steps; and (2) The diffusion module receives as input an already complete and physically reasonable macromolecular structure, rather than random noise as in the training regimes of *AlphaFold3* or *Boltz-2* for de novo structure prediction.

These findings confirm that for the specific task of structural refinement against density maps, a targeted one-step diffusion scheme is far superior to classical multi-step generation.

## F.3 ABLATION STUDY

We performed an ablation study on 27 protein complexes (Table 5). The full *CryoNet.Refine* configuration achieves the best overall performance between model-map correlation coefficients and model geometric metrics, underscoring the necessity of integrating density- and geometry-aware objectives.

Table 5: Ablation study: how loss functions influence model-map correlation coefficients and model geometric metrics

| Category | Metrics | $\gamma_{\mathrm{den}} = 0$ | $\gamma_{\mathrm{rama}} = 0$ | $\gamma_{\mathrm{rot}} = 0$ | *CryoNet.Refine* |
|---|---|---|---|---|---|
| Model-map Correlation Coefficients | $CC_{\mathrm{mask}} \uparrow$ | 0.41 | **0.65** | 0.64 | **0.65** |
| | $CC_{\mathrm{box}} \uparrow$ | 0.42 | **0.58** | 0.57 | **0.58** |
| | $CC_{\mathrm{mc}} \uparrow$ | 0.42 | **0.66** | 0.65 | **0.66** |
| | $CC_{\mathrm{sc}} \uparrow$ | 0.41 | **0.64** | **0.64** | 0.63 |
| | $CC_{\mathrm{peaks}} \uparrow$ | 0.24 | **0.48** | 0.47 | 0.47 |
| | $CC_{\mathrm{volume}} \uparrow$ | 0.43 | **0.65** | 0.64 | **0.65** |
| Model Geometric Metrics | Angle RMSD (°)↓ | 0.45 | **0.41** | 0.44 | 0.54 |
| | C-beta deviations↓ | 0.00 | 0.00 | 0.00 | 0.00 |
| | Ramachandran favored (%)↑ | 99.09 | 90.75 | **99.22** | 98.80 |
| | Ramachandran outlier (%)↓ | 0.06 | 2.27 | **0.03** | 0.10 |
| | Rotamer favored (%)↑ | **98.67** | 98.64 | 94.48 | 98.58 |
| | Rotamer outlier (%)↓ | 0.54 | 2.11 | 1.38 | **0.51** |

Removing the density loss ($\gamma_{\mathrm{den}} = 0$) leads to a pronounced drop across all CC metrics, with $CC_{\mathrm{mask}}$ and $CC_{\mathrm{mc}}$ reduced by over 35%, confirming that the density term is indispensable for guiding accurate model–map fitting. In contrast, omitting the Ramachandran prior ($\gamma_{\mathrm{rama}} = 0$) preserves correlation coefficients but severely compromises geometry, reducing favored residues from 98.80% to 90.75% and inflating outliers more than 20-fold. This highlights the critical role of stereochemical priors in constraining backbone conformations.

The effect of removing the rotamer prior ($\gamma_{\mathrm{rot}} = 0$) is more nuanced: CC values remain competitive, and Ramachandran statistics are slightly improved, but side-chain packing deteriorates, with favored rotamers dropping to 94.48%. This indicates that local side-chain restraints complement backbone priors in ensuring chemically realistic conformations.

Together, these results establish three key principles: (i) density loss is essential for achieving high model–map correlation, (ii) stereochemical priors, particularly Ramachandran constraints, safeguard backbone geometry, and (iii) rotamer priors contribute to realistic side-chain packing. The synergy of these components underpins *CryoNet.Refine*'s ability to deliver models that are both density-consistent and stereochemically robust.

## F.4 COMPARISON WITH DIRECT NUMERICAL OPTIMIZATION

To investigate whether the generative capabilities of our diffusion model are strictly necessary, or if the structural improvements are merely due to the proposed differentiable loss functions, we implemented a **pure numerical coordinate optimization baseline**. This baseline involves no neural networks, no diffusion process, and no learned structural priors.

**Experimental Setup.** In this baseline, we treated the atomic coordinates of the initial *AlphaFold3* prediction as learnable parameters ($\mathbf{x} \in \mathbb{R}^{3 \times N_{\mathrm{atoms}}}$). The optimization process was designed to be as close as possible to the CryoNet.Refine inference loop:

- **Inputs:** The initial *AlphaFold3* structure and the experimental cryo-EM density map.

- **Objective Function:** We utilized the identical composite loss function used in Cryo-Net.Refine: $\mathcal{L} = \mathcal{L}_{\text{den}} + \mathcal{L}_{\text{geo}}$, employing the exact same weights ($\gamma$) for all density and geometric terms.

- **Optimization:** The coordinates were updated directly via backpropagation using a **Stochastic Gradient Descent (SGD)** optimizer with a **momentum of 0.9**.

- **Schedule:** To ensure a fair comparison with our diffusion model, we set the maximum number of iterations (recycles) to 300, matching the diffusion inference budget. An early stopping mechanism with a patience of 20 iterations was implemented, along with a `ReduceLROnPlateau` scheduler to dynamically adjust the learning rate.

**Results.** We evaluated this baseline on a representative subset of 27 protein complexes (Table 6). The results reveal a critical limitation of pure numerical optimization. While the baseline achieves excellent geometric scores (e.g., the lowest Angle RMSD of 0.27°), it **fails to effectively fit the density map**, yielding a $CC_{\text{mask}}$ of only 0.46, which is marginally better than the initial input (0.44) but significantly worse than CryoNet.Refine (0.65).

This indicates that without the generative and exploratory capabilities of the diffusion model, the gradient descent process gets trapped in local minima close to the initial structure. It satisfies the geometric constraints (which are local) but cannot traverse the energy landscape to find the global conformation that matches the experimental density. Thus, the **one-step diffusion module is essential** for balancing data fidelity with biological plausibility.

Table 6: Comparison between CryoNet.Refine and Direct Numerical Optimization (SGD) on a subset of 27 protein complexes. The numerical baseline optimizes geometry well but fails to fit the density map.

| Category | Metrics | AlphaFold3 | Phenix. real_space_refine | Numerical Optimization | CryoNet. Refine |
|---|---|---|---|---|---|
| Model-map Correlation Coefficients | $CC_{\text{mask}} \uparrow$ | 0.44 | 0.61 | 0.46 | **0.65** |
| | $CC_{\text{box}} \uparrow$ | 0.44 | 0.55 | 0.47 | **0.58** |
| | $CC_{\text{mc}} \uparrow$ | 0.45 | 0.62 | 0.46 | **0.66** |
| | $CC_{\text{sc}} \uparrow$ | 0.44 | 0.61 | 0.45 | **0.64** |
| | $CC_{\text{peaks}} \uparrow$ | 0.31 | 0.44 | 0.32 | **0.47** |
| | $CC_{\text{volume}} \uparrow$ | 0.47 | 0.61 | 0.48 | **0.65** |
| Model Geometric Metrics | Angle RMSD (°)↓ | 1.60 | 0.73 | **0.27** | 0.54 |
| | $C_\beta$ deviations↓ | 0.09 | **0.00** | **0.00** | **0.00** |
| | Ramachandran favored (%)↑ | 95.48 | 96.33 | 98.03 | **99.68** |
| | Ramachandran outlier (%)↓ | 1.04 | **0.05** | 0.22 | 0.11 |
| | Rotamer favored (%)↑ | 96.80 | 80.99 | 97.43 | **98.58** |
| | Rotamer outlier (%)↓ | 1.14 | 1.48 | 0.87 | **0.51** |

## F.5 RUNTIME PROFILING

To provide deeper insights into the computational efficiency of *CryoNet.Refine*, we conducted a detailed runtime profile analysis.

We profiled the execution time of a single refinement recycle for a representative protein complex (e.g., PDB-6ksw, ∼1,900 residues). The breakdown of time consumption per iteration is detailed in Table 7.

The results indicate that the Gradient Computation (Backpropagation) and the Loss Calculation are the two primary computational bottlenecks. The forward pass of the diffusion model is relatively efficient due to the one-step sampling.

Table 7: Runtime profile of a single refinement recycle (averaged over total recycle).

| Component | Time (ms) | Percentage (%) |
|---|---|---|
| Data Loading & Preprocessing | 150 | ∼6% |
| Model Forward Pass (One-step) | 200 | ∼7% |
| Loss Calculation (Density + Geometry) | 940 | ∼35% |
| Backpropagation & Optimization | 1420 | ∼52% |
| **Total per Recycle** | **2710** | **100%** |

