# OpenReview forum: "CryoNet.Refine: A One-step Diffusion Model for Rapid Refinement of Structural Models with Cryo-EM Density Map Restraints"
_ICLR.cc/2026/Conference — ICLR 2026 Poster_

### Official Review · Reviewer_yjst · 2025-10-26

**Soundness:** 2
**Presentation:** 3
**Contribution:** 2
**Rating:** 4
**Confidence:** 4

**Summary:**

This paper proposes CryoNet.Refine, a neural refinement framework that integrates a one-step diffusion model with experimental cryo-EM density map restraints.
Unlike traditional refinement programs such as Phenix.real_space_refine, CryoNet.Refine introduces:
1. A learnable density model that maps atomic coordinates to voxelized densities, enabling end-to-end differentiable correlation with experimental maps, and

2. A geometry loss suite that includes differentiable Ramachandran, rotamer, and bond-length/angle penalties.

The model refines initial structures (often AlphaFold3 predictions) toward experimental densities, showing improvements in map correlation (CCmask, CCmain-chain) and stereochemical quality across 63 cryo-EM complexes.

**Strengths:**

- The combination of a density-aware differentiable loss and stereochemical constraints is technically sound.
- The paper is well written
- The paper demonstrates consistent gains over Phenix in both correlation coefficients and geometric metrics (notably, Ramachandran favored and rotamer outliers).

**Weaknesses:**

- The proposed “one-step diffusion” formulation is described as an innovation, but its necessity is not clearly demonstrated. The ablation shows marginal benefit, and it is unclear why a diffusion step—traditionally for generative sampling—improves deterministic refinement. A comparison to standard coordinate optimization (e.g., gradient descent using the same losses) would help establish that the diffusion mechanism contributes beyond architecture novelty.
- The “density generator” is mentioned but not architecturally described (layers, parameters, training regime, loss balance weights γ). It is also unclear how the network differs from a Gaussian scattering baseline (molmap from ChimeraX).
- The ablation in Table 4 shows individual loss terms but does not isolate the impact of the density model vs. geometry components.
The diffusion vs. non-diffusion refinement comparison (one-step vs. multi-step) is qualitative; quantitative runtime and accuracy tradeoffs are missing.

**Questions:**

1. What concrete advantage does the diffusion step provide compared to standard coordinate optimization using the same losses?
2. The authors state that they developed a novel, parameter-free, and differentiable density generator in the Introduction, but Section 3.2.1 indicates that they simply follow the ‘molmap’ implementation in ChimeraX. What, specifically, is novel in this component?

**Details Of Ethics Concerns:**

No concerns

---

> ### Author Response · Authors · 2025-11-19
> **Response to Reviewer yjst (Weaknesses)**
>
> ***Weaknesses:***
>
> **W1**: The proposed “one-step diffusion” formulation is described as an innovation, but its necessity is not clearly demonstrated. The ablation shows marginal benefit, and it is unclear why a diffusion step—traditionally for generative sampling—improves deterministic refinement. A comparison to standard coordinate optimization (e.g., gradient descent using the same losses) would help establish that the diffusion mechanism contributes beyond architecture novelty.
>
> **Response W1**: Following your advice, we implemented a numerical optimization baseline starting from the initial structure that directly optimizes atomic structure coordinates using gradient descent on the same loss functions (density map + geometric restraints) without any neural network components.
>
> Method Comparison Table (N=27)
>
> |Category|Metrics|*AlphaFold3*|*Phenix.real_space_refine*|Numerical Optimization|*CryoNet.Refine*|
> |--|--|--|--|--|--|
> |**Model–map Correlation**|$\rm CC_{mask}\uparrow$|0.44|0.61|0.46|**0.65**|
> ||$\rm CC_{box}\uparrow$|0.44|0.55|0.47|**0.58**|
> ||$\rm CC_{mc}\uparrow$|0.45|0.62|0.46|**0.66**|
> ||$\rm CC_{sc}\uparrow$|0.44|0.61|0.45|**0.64**|
> ||$\rm CC_{peaks}\uparrow$|0.31|0.44|0.32|**0.47**|
> ||$\rm CC_{volume}\uparrow$|0.47|0.61|0.48|**0.65**|
> |**Model Geometric Metrics**|Angle RMSD (degree)$\downarrow$|1.60|0.73|**0.27**|0.54|
> ||$C_\beta$ deviations$\downarrow$|0.09|**0.00**|**0.00**|**0.00**|
> ||Ramachandran favored (%)$\uparrow$|95.48|96.33|98.03|**99.68**|
> ||Ramachandran outlier (%)$\downarrow$|1.04|**0.05**|0.22|0.11|
> ||Rotamer favored (%)$\uparrow$|96.8|80.99|97.43|**98.58**|
> ||Rotamer outlier (%)$\downarrow$|1.14|1.48|0.87|**0.51**|
>
> > Performance on protein complex (↑: higher is better, ↓: lower is better).
>
> The results show that our CryoNet.Refine outperforms numerical optimization method both on model-map correlation coefficients and model geometric metrics, which highlights the critical role of our diffusion module (initialized with Boltz-2, a reproduction of AlphaFold3 (Abramson et al., Nature 2024). This module was pre-trained on over 270K macromolecular structures, learning rich priors about natural protein structures and the underlying rules, which is entirely absent in a simple numerical optimization scheme. It's one of the major factors why our method outperforms the baseline described above.
>
> **W2**: The "density generator" is mentioned but not architecturally described (layers, parameters, training regime, loss balance weights γ). It is also unclear how the network differs from a Gaussian scattering baseline (molmap from ChimeraX).
>
> **Response W2**: Our density generator conceptually follows the "molmap" method in ChimeraX, which is a physics-based simulator that generates volume maps by summing Gaussian functions centered on atomic structure positionscoordinates. The key innovation is that we implemented this procedure using PyTorch to be fully differentiable, unlike molmap that relies on non-differentiable libraries (like NumPy). It allows the backpropagation of density loss into the diffusion module as a crucial guidance to fit experimental cryo-EM density maps during refinement .(Algorithm details are in Section 3.2.1).
>
> **W3**: The ablation in Table 4 shows individual loss terms but does not isolate the impact of the density model vs. geometry components. The diffusion vs. non-diffusion refinement comparison (one-step vs. multi-step) is qualitative; quantitative runtime and accuracy tradeoffs are missing.
>
> **Response W3**：We have expanded the original ablation study to include a quantitative experiment comparing our one-step diffusion approach with multi-step strategy across the 27 cases. The results, presented in the table below, further demonstrate the rational of our one-step strategy.
>
> |Category|Metrics|*CryoNet.Refine*|*Classical Diffusion (200 steps)*|
> |--|--|--|--|
> |**Model–map Correlation**|$\rm CC_{mask}\uparrow$|**0.65**|0.20|
> ||$\rm CC_{box}\uparrow$|**0.58**|0.28|
> ||$\rm CC_{mc}\uparrow$|**0.66**|0.21|
> ||$\rm CC_{sc}\uparrow$|**0.63**|0.21|
> ||$\rm CC_{peaks}\uparrow$|**0.47**|0.13|
> ||$\rm CC_{volume}\uparrow$|**0.65**|0.27|
> |**Model Geometric Metrics**|Angle RMSD (degree)$\downarrow$|**0.54**|9.80|
> ||$C_\beta$ deviations$\downarrow$|**0.00**|33.04|
> ||Ramachandran favored (%)$\uparrow$|**98.80**|74.01|
> ||Ramachandran outlier (%)$\downarrow$|**0.10**|12.39|
> ||Rotamer favored (%)$\uparrow$|**98.58**|81.86|
> ||Rotamer outlier (%)$\downarrow$|**0.51**|9.21|

---

> > ### Comment · Reviewer_yjst · 2025-11-25
> >
> > Thanks for the authors' answers. Could you clarify the details of the experimental setup for ‘Response 1’ and ‘Response 3’?

---

> > > ### Author Response · Authors · 2025-11-26
> > > **Response to Reviewer yjst**
> > >
> > > Q: Thanks for the authors' answers. Could you clarify the details of the experimental setup for ‘Response 1’ and ‘Response 3’?
> > >
> > > **Response 1 — Experimental Setup in W1:**
> > >
> > > Below we clarify the exact experimental setup used for the Numerical Optimization baseline in Response W1.
> > >
> > > To isolate the value of the diffusion module, we constructed a **pure numerical coordinate optimization** , with **no neural networks** , **no diffusion** , and **no learned priors** .
> > >
> > > 1. **Inputs**: AF3 structure + experimental cryo-EM map (same as CryoNet.Refine)
> > > 2. **Optimization object**: Atomic coordinates treated as a learnable torch.Parameter
> > > 3. **Loss functions**: Identical to CryoNet.Refine (density loss + geometric restraints, same weights γ)
> > > 4. **Update rule**: Backpropagation directly updates coordinates using SGD (momentum=0.9)
> > > 5. **Training schedule** (matched to CryoNet.Refine for fairness):
> > >     - 300 recycles
> > >     - At each recycle: compute loss → sum → backprop → SGD update
> > >     - Early stopping patience = 20
> > >
> > > The following minimal optimizer was implemented:
> > > ```python
> > > self.learnable_coords = torch.nn.Parameter(initial_coords.requires_grad_(True))
> > > self.optimizer = torch.optim.SGD(
> > >     [self.learnable_coords],
> > >     lr=self.refine_args.learning_rate,
> > >     momentum=0.9
> > > )
> > > self.scheduler = ReduceLROnPlateau(...)
> > > ```
> > > This experiment represents the strongest possible non-neural, non-diffusion baseline; any performance gap thus reflects the benefit of learned structural priors in our diffusion module.
> > >
> > >
> > > **Response 2 — Experimental Setup in W3**
> > >
> > > To ensure a fair comparison with the classical 200-step diffusion method, we utilized the identical dataset (encompassing both the training and evaluation sets), the same loss functions, and ensured that both diffusion processes start from the same initial atomic coordinates. The key difference is that, unlike our One-step Diffusion model, the classical 200-step diffusion leverages the original AlphaFold3 DDPM hyperparameters and noise schedule (performing all 200 steps within a single recycle).
> > >
> > > This setup isolates diffusion strategy rather than training data, loss functions, or input conditions, showing that our one-step formulation is substantially more stable and effective than directly applying a classical multi-step diffusion chain during refinement.

---

> > > > ### Comment · Reviewer_yjst · 2025-11-27
> > > >
> > > > Thank you for the detailed response. The numerical optimization baseline (Response W1) clearly isolates the contribution of learned structural priors. I have a follow-up regarding the Classical Diffusion baseline (Response W3). The reported performance is extremely poor ($CC_{mask}=0.20$, Angle RMSD=9.80), which suggests the structure was severely degraded. Could you clarify the specific noise schedule and initialization used for this baseline? Did you apply the full noise schedule ($t=T$) to the input structure, or did you use a partial noise schedule (starting from $t < T$)?

---

> > > > > ### Author Response · Authors · 2025-11-28
> > > > > **Response to Reviewer yjst**
> > > > >
> > > > > **Comment:**
> > > > >
> > > > > Thank you for the detailed response. The numerical optimization baseline (Response W1) clearly isolates the contribution of learned structural priors. I have a follow-up regarding the Classical Diffusion baseline (Response W3). The reported performance is extremely poor ($CC_{mask}=0.20$, Angle RMSD=9.80), which suggests the structure was severely degraded. Could you clarify the specific noise schedule and initialization used for this baseline? Did you apply the full noise schedule ($t=T$) to the input structure, or did you use a partial noise schedule (starting from$t<T$ )?
> > > > >
> > > > > **Response:**
> > > > >
> > > > > Thank you for the follow-up question. For the Classical Diffusion baseline (Response W3), we confirm that we used the full 200-step AlphaFold3 noise schedule with all hyperparameters unchanged (num_sampling_steps = 200, σ_min=0.0004, σ_max=160, ρ=7, etc.). The sampling schedule implementation is identical to AlphaFold3.
> > > > >
> > > > > The differences between our  'Classical Diffusion (200 steps + normalized sigmas)' and AlphaFold3 are:
> > > > >
> > > > > 1. The diffusion sampling starts from the input structure (instead of Gaussian noise)
> > > > > 2. We applied a normalization to the sigma schedule .
> > > > >
> > > > > We observed that this normalization leads to significant structural collapse during multi-step sampling. To validate this finding, we tested this by testing the baseline without normalization. As expected, the stability was improved, yet the overall performance remained below the results achieved by our approach.
> > > > >
> > > > > **Summary of results:**
> > > > >
> > > > > |Category|Metrics|||CryoNet.Refine|Classical Diffusion (200 steps + normalized sigmas)|Classical Diffusion (200 steps)|
> > > > > |--|--|--|--|--|--|--|
> > > > > |**Model–map Correlation**|CC_mask ↑|||0.65|0.20|0.30|
> > > > > ||CC_box ↑|||0.58|0.28|0.35|
> > > > > ||CC_mc ↑|||0.66|0.21|0.31|
> > > > > ||CC_sc ↑|||0.63|0.21|0.31|
> > > > > ||CC_peaks ↑|||0.47|0.13|0.24|
> > > > > ||CC_volume ↑|||0.65|0.27|0.34|
> > > > > |**Model Geometry**|Angle RMSD (°) ↓|||0.54|9.80|1.66|
> > > > > ||Cβ deviations ↓|||0.00|33.04|0.0041|
> > > > > ||Ramachandran favored (%) ↑|||98.80|74.01|98.63|
> > > > > ||Ramachandran outlier (%) ↓|||0.10|12.39|0.18|
> > > > > ||Rotamer favored (%) ↑|||98.58|81.86|98.63|
> > > > > ||Rotamer outlier (%) ↓|||0.51|9.21|0.18|
> > > > >
> > > > > These results further highlight that our one-step diffusion refinement is fundamentally more suitable for cryo-EM atomic structure refinement than classical multi-step generative diffusion.

---

> ### Author Response · Authors · 2025-11-19
> **Response to Reviewer yjst (Questions)**
>
> **Questions**:
>
> **Q1**. What concrete advantage does the diffusion step provide compared to standard coordinate optimization using the same losses?
>
> **Response Q1**：The results (table in Response W1) show that our CryoNet.Refine outperforms numerical optimization method both on model-map correlation coefficients and model geometric metrics, which highlights the critical role of our diffusion module (initialized with Boltz-2, a reproduction of AlphaFold3 (Abramson et al., Nature 2024). This module was pre-trained on over 270K macromolecular structures, learning rich priors about natural protein structures and the underlying rules, which is entirely absent in a simple numerical optimization scheme. It's one of the major factors why our method outperforms the baseline described above.
>
> **Q2**.The authors state that they developed a novel, parameter-free, and differentiable density generator in the Introduction, but Section 3.2.1 indicates that they simply follow the ‘molmap’ implementation in ChimeraX. What, specifically, is novel in this component?
>
> **Response Q2**: Our density generator conceptually follows the "molmap" method in ChimeraX, which is a physics-based simulator that generates volume maps by summing Gaussian functions centered on atomic structure coordinates. The key innovation is that we implemented this procedure using PyTorch to be fully differentiable, unlike molmap that relies on non-differentiable libraries (like NumPy). It allows the backpropagation of density loss into the diffusion module as a crucial guidance to fit experimental cryo-EM density maps during refinement (Algorithm details are in Section 3.2.1).

---

### Official Review · Reviewer_aP6u · 2025-10-27

**Soundness:** 3
**Presentation:** 3
**Contribution:** 3
**Rating:** 6
**Confidence:** 3

**Summary:**

CryoNet.Refine is an end-to-end framework that automates and accelerates molecular structure refinement. It utilizes a one-step diffusion model that integrates a density-aware loss function with robust stereochemical restraints, enabling it to rapidly optimize a structure against the experimental data.

**Strengths:**

- This study bridges the gap between structure prediction models and cryoEM densities in a modern approach. With some carefully designed loss functions, CryoNet.Refine brings the power of folding models to cryo-EM model building.
- The ablation study is comprehensive, and the figures are well made.

**Weaknesses:**

- Why is the model named “one-step diffusion” but takes several recycling numbers?
- The method part lacks several technical details, and is hard to follow.
	- What is the training set of CryoNet.Refine? Is it “trained” for each protein (like ReLION/CryoDRGN), or trained over a set of proteins and evaluated on some test set without tuning the model parameters?
    - In Section 3.1, line 202, the authors wrote that the model is initialized from Boltz-2’s parameters.
		- Does CryoNet.Refine has exactly the same model architecture as Boltz-2?
		- If I am understanding correctly, CryoNet.Refine can be viewed as something like AF3 with classifier guidance. Is that true?
	- In line 208, the authors said that AF3 requires hundreds of sampling steps. In line 212, the authors wrote that one-step diffusion poses a key advantage that guidance can be performed directly on the predictions. I admit these two statements, but what confused me is that: AF3 is also a diffusion model which predicts the final sample (instead of predicting the velocity or noise), what is the difference between CryoNet.Refine and AF3?
		- Can I view AF3 as a one-step diffusion model, although it takes hundreds of steps in the sampling process?

**Questions:**

- In Section 3.1, line 193, the initial atomic structure is fed into CryoNet.Refine. You said that the model derived pair representation $z$ from the atomic structures. What is the shape of the pair representation? Is the side length of $z$ the number of residues, or the number of atoms?
- I think AF3 can also benefit from the loss you proposed. What if adopting the loss you proposed to the predicted sample of AF3, like diffusion posterior sampling? (ref: (1) Diffusion posterior sampling for general noisy inverse problems, (2) CryoFM: a flow-based foundation model for cryoEM densities)
- Since the model is initialzied from Boltz-2, why do you need some geometry loss to constrain the structures? I think a folding model has already contained such knowledge.

---

> ### Author Response · Authors · 2025-11-19
> **Response to Reviewer aP6u (Weaknesses)**
>
> **Weaknesses:**
>
>
> **W1:** Why is the model named “one-step diffusion” but takes several recycling numbers?
>
>
> **Response W1**：Thank you for your question that highlights two distinct "steps" in our methodology. The apparent contradiction arises from the fact that "step" refers to two different concepts:
>
>
> 1. The "One-step" refers to the diffusion sampling algorithm: In our work, we employ a deterministic diffusion sampler (based on pre-conditioned network). Unlike stochastic sampling that requires hundreds or thousands of iterations, our sampler can, in a single forward pass (one step), map a noisy structure $x_t$ directly to a predicted clean structure $x_0$, given a noise level σ. This makes the core diffusion process extremely efficient.
>
> 2. The "Recycling" refers to the overall refinement loop: For high-precision tasks like structural refinement, a single "one-step" sampling may not be sufficient to reach the global optimum. Therefore, we introduce an outer, multi-round optimization loop. In each recycle:
>
> We perform another "one-step diffusion" forward pass. Then, we compute the density and geometry losses and perform backpropagation to update the model parameters, customizing it for the current specific case.
>
>
> In summary, the name "One-step Diffusion" describes the efficiency of our core sampler, while "Multiple Recycles" describes the optimization strategy. This combination enables the diffusion module to progressively converge to the most accurate structure that fits the input experimental data.
>
>
> **W2.1:**  The method part lacks several technical details, and is hard to follow. What is the training set of CryoNet.Refine? Is it  “trained” for each protein (like ReLION/CryoDRGN), or trained over a set of proteins and evaluated on some test set without tuning the model parameters?
>
>
> **Response W2.1:** Yes! CryoNet.Refine is trained for each protein like CryoDRGN.
>
>
> **W2.2**: In Section 3.1, line 202, the authors wrote that the model is initialized from Boltz-2’s parameters. Does CryoNet.Refine has exactly the same model architecture as Boltz-2?
>
>
> **Response W2.2**：The direct answer is No, CryoNet.Refine has a substantially different architecture from Boltz-2 (AlphaFold3). While several components of CryoNet.Refine - including template module, pairformer module, and diffusion module - are initialized from the pretrained weights of Boltz-2 (as noted in Section 3.1, line 202), our architecture incorporates two major modifications: removal of the MSA module, and the one-step diffusion sampler, both are tailored specifically for efficient, density map restrained structure refinement.
>
>
> **W2.3**: If I am understanding correctly, CryoNet.Refine can be viewed as something like AF3 with classifier guidance. Is that true?
>
>
> **Response W2.3**：It's important to note that the underlying paradigms are fundamentally different. CryoNet.Refine performs a refinement task (structure input, structure output) that uses the differentiable density and geometry losses to update the diffusion module's parameters for the specific input structure, differing from AF3's noise-to-structure generation (1D-sequence input, structure output).
>
> **W2.4**: In line 208, the authors said that AF3 requires hundreds of sampling steps. In line 212, the authors wrote that one-step diffusion poses a key advantage that guidance can be performed directly on the predictions. I admit these two statements, but what confused me is that: AF3 is also a diffusion model which predicts the final sample (instead of predicting the velocity or noise), what is the difference between CryoNet.Refine and AF3?
>
>
> **Response W2.4**：The primary difference lies in the starting point for diffusion sampling: AF3 starts from random noise, while CryoNet.Refine starts from the input atomic structure. This allows CryoNet.Refine to bypass lengthy denoising and perform guided refinement in a single step (within several refinement loops). Essentially, CryoNet.Refine is not a de novo structure predictor but a refinement tool that improves a given structure's geometry quality and its fit to the cryo-EM density.
>
>
> **W2.5**: Can I view AF3 as a one-step diffusion model, although it takes hundreds of steps in the sampling process?
>
>
> **Response W2.5**：No. In this context, AF3 can be viewed as a 200-step diffusion model.

---

> ### Author Response · Authors · 2025-11-19
> **Response to Reviewer aP6u (Questions)**
>
> ***Questions:***
>
> **Q1**: In Section 3.1, line 193, the initial atomic structure is fed into CryoNet.Refine. You said that the model derived pair representation from the atomic structures. What is the shape of the pair representation? Is the side length of the number of residues, or the number of atoms?
>
> **Response Q1**：Yes, it's the side length of the number of residues. The shape of the pair representation is [n, n, c], where "n" is the number of residues and "c" is the size of feature channels.
>
> **Q2**: I think AF3 can also benefit from the loss you proposed. What if adopting the loss you proposed to the predicted sample of AF3, like diffusion posterior sampling? (ref: (1) Diffusion posterior sampling for general noisy inverse problems, (2) CryoFM: a flow-based foundation model for cryoEM densities)
>
> **Response Q2**: Thanks for your highly insightful suggestion that points toward a new paradigm for experimental integration. Yes — our losses (expecially for the ramachandran loss, rotamer loss, and density loss) can indeed be integrated into an AF3-like framework. As stated in line 481, our density and geometry losses are model-agnostic and can serve as effective guidance signals. Incorporating them for conditional diffusion sampling would directly steer AF3’s generative trajectory toward both density consistency and stereochemical accuracy. This is a promising direction fully aligned with our design.
>
> **Q3**: Since the model is initialzied from Boltz-2, why do you need some geometry loss to constrain the structures? I think a folding model has already contained such knowledge.
>
> **Response Q3**：Exactly! The folding model indeed has already contained geometry priors, but this is insufficient for the atomic-level precision required for high-resolution refinement. Our method addresses this by introducing explicit, differentiable density and geometry losses during the training loop to enforce local stereochemical rules and strict alignment with the experimental map.

---

### Official Review · Reviewer_6iZo · 2025-10-28

**Soundness:** 3
**Presentation:** 2
**Contribution:** 2
**Rating:** 6
**Confidence:** 3

**Summary:**

This paper proposed a model for the refinement of atomic models based on CryoEM density map. Specifically, this work finetunes Boltz2 for each pair of density map and atomic model with the density loss and the geometry loss. The structure is output by a specially designed one-step diffusion module which enables direct back-propagation from the complex loss functions.

**Strengths:**

- The density loss is conceptually elegant and novel. The one-step diffusion module is well motivated as typical diffusion models can only be trained using specific loss function. The one-step diffusion allows flexible loss defitions.
- Structure-specific post-training of the Boltz2 model ensures generalization. Previous works directly learn a mapping from density maps to atomic models, which might fail on structures that are significantly different from training data.
- Better performance than previous methods.

**Weaknesses:**

- As it requires fine-tuning for each structure at inference time, the efficiency is still limited, and the efficiency improvement is not very significant compared to previous methods.

**Questions:**

- What would happen if we directly optimize the coordinates of the input structure instead of network parameters using the density loss and the geometry loss?

---

> ### Author Response · Authors · 2025-11-19
> **Response to Reviewer  6iZo**
>
> **W1.** As it requires fine-tuning for each structure at inference time, the efficiency is still limited, and the efficiency improvement is not very significant compared to previous methods.
>
> **Response W1:**
> We acknowledge this limitation. Our per-structure optimization approach prioritizes refinement quality over speed. We recognize efficiency as an important factor and we plan to explore in future work: (1) faster convergence strategies (2) parallel refinement (3) more efficient loss computation .
>
> **Q1.** What would happen if we directly optimize the coordinates of the input structure instead of network parameters using the density loss and the geometry loss?
>
> **Response Q1:**
> Following your advice, we implemented a numerical optimization baseline where atomic structure coordinates are treated as learnable parameters and optimized via SGD with the same loss functions (density map + geometric restraints). This approach completely bypasses neural network components—no diffusion module or structural encoders—providing a pure gradient-descent baseline that directly updates atomic structure coordinates.
>
> > Method Comparison Table (N=27)
>
> |Category|Metrics|*AlphaFold3*|*Phenix.real_space_refine*|*Numerical Optimization*|*CryoNet.Refine*|
> |--|--|--|--|--|--|
> |**Model–map Correlation**|$\rm CC_{mask}\uparrow$|0.44|0.61|0.46|**0.65**|
> ||$\rm CC_{box}\uparrow$|0.44|0.55|0.47|**0.58**|
> ||$\rm CC_{mc}\uparrow$|0.45|0.62|0.46|**0.66**|
> ||$\rm CC_{sc}\uparrow$|0.44|0.61|0.45|**0.64**|
> ||$\rm CC_{peaks}\uparrow$|0.31|0.44|0.32|**0.47**|
> ||$\rm CC_{volume}\uparrow$|0.47|0.61|0.48|**0.65**|
> |**Model Geometric Metrics**|Angle RMSD (degree)$\downarrow$|1.60|0.73|**0.27**|0.54|
> ||$C_\beta$ deviations$\downarrow$|0.09|**0.00**|**0.00**|**0.00**|
> ||Ramachandran favored (%)$\uparrow$|95.48|96.33|98.03|**99.68**|
> ||Ramachandran outlier (%)$\downarrow$|1.04|**0.05**|0.22|0.11|
> ||Rotamer favored (%)$\uparrow$|96.8|80.99|97.43|**98.58**|
> ||Rotamer outlier (%)$\downarrow$|1.14|1.48|0.87|**0.51**|
>
> > Performance on protein complex (↑: higher is better, ↓: lower is better).
>
> The results above show that our CryoNet.Refine outperforms numerical optimization method both on model-map correlation coefficients and model geometric metrics, which highlights the critical role of our diffusion module (initialized with Boltz-2, a reproduction of AlphaFold3 (Abramson et al., Nature 2024). This module was pre-trained on over 270K macromolecular structures, learning rich priors about natural protein structures and the underlying rules, which is entirely absent in a simple numerical optimization scheme. It's one of the major factors why our method outperforms the baseline described above.

---

### Official Review · Reviewer_9SBf · 2025-10-31

**Soundness:** 3
**Presentation:** 2
**Contribution:** 3
**Rating:** 6
**Confidence:** 3

**Summary:**

This paper focuses on the process of fitting atomic models into experimental density maps for structure determination by cryo-electron microscopy (cryo-EM). Traditional refinement pipelines are computationally expensive and require extensive manual tuning. To address these challenges, the authors present CryoNet.Refine, an end-to-end deep learning framework that automates and accelerates molecular structure refinement. Specifically, CryoNet.Refine employs a one-step diffusion model that integrates a density-aware loss function with robust stereochemical restraints to rapidly optimize structures against experimental data. The framework is capable of refining not only protein complexes but also DNA/RNA–protein assemblies. Experimental results demonstrate that CryoNet.Refine achieves clear improvements over traditional approaches in both model–map correlation and overall geometric accuracy.

**Strengths:**

1. This paper connects two important things in structural biology, computational modeling (e.g., AlphaFold3) and cryo-EM experimental density maps.

2. The architecture design is well-motivated, improving efficiency while maintaining refinement quality.

3. Presented experimental results demonstrate strong refinement performance, reduced manual effort, and clear efficiency gains over traditional methods.

**Weaknesses:**

1. It is confusing what exactly happens during a training step versus inference. I assume Fig. 2 provides an overview of the training process. During inference, there seems to be no computation of density or geometry loss, and the model likely performs only a single pass through the Atom Encoder, Sequence Embedder, and Diffusion Module. Clarifying this distinction would help readers better understand the workflow and computational efficiency.

2. Several simple yet informative baselines are missing — for example, numerical optimization starting from the initial structure. Including such comparisons would better contextualize the claimed improvements.

3. There is no ablation study comparing the classical diffusion model with the one-step diffusion approach.

4. The density generator component is not evaluated. Details about its data patterns or visualizations are absent, and there is no discussion on how well the generated densities align with experimental cryo-EM maps, which can vary significantly depending on the instrument and imaging conditions.

5. The number of reported test cases is too small. A larger-scale evaluation is necessary to establish robustness and generalizability across diverse molecular systems.

**Questions:**

1. If my assumption in Weakness 1 is correct, that no density or geometry loss is computed during inference, would it be beneficial to incorporate these losses as a form of test-time refinement?

2. Figure 2 includes a Pairformer module, but it is never discussed in the text. Is this a Transformer-based component where cross-attention occurs between atom and sequence embeddings?

---

> ### Author Response · Authors · 2025-11-19
> **Response to Reviewer 9SBf (Weaknesses 4~5 & Questions 1~2 )**
>
> **W4.** The density generator component is not evaluated. Details about its data patterns or visualizations are absent, and there is no discussion on how well the generated densities align with experimental cryo-EM maps, which can vary significantly depending on the instrument and imaging conditions.
>
> **Response W4** In response to your suggestion, We add a figure to show the data patterns and visualizations of the density generator in Appendix D. The density generator is not a neural network but a physics-based simulator. It constructs the target density volume by summing Gaussian functions centered on atomic structure coordinates with the simulation process specifically tailored to the target resolution. The main methodological contribution regarding density lies in our introduction of a fully differentiable density loss, which serves as guidance signal during the generative process.
> To evaluate how well the simulated maps align with input experimental maps, we calculate the correlation coefficients across all cases. The average correlation is 0.749 and over 80% of all cases exceed 0.70. Nevertheless, we agree with the reviewer that simulated maps fail to capture the complex characteristics arising from diverse experimental instruments and imaging conditions. They represent the "ideal" density distribution without experimental artifacts, and provide ground truth for what high-frequency features should look like. Currently, no existing methods could address this experimental artifacts in cryo-EM community. We are considering replacing the current simulation-based density generator with a deep learning–based generative model, which would better reproduce the statistical and structural features of experimental cryo-EM maps.
>
> **W5.** The number of reported test cases is too small. A larger-scale evaluation is necessary to establish robustness and generalizability across diverse molecular systems.
>
> **Response W5:** We thank the reviewer for raising the important point regarding evaluation benchmark scale. We performed further experiments on a larger test set, expanding from 53 cases to 110 protein complexes. The results, which will be fully incorporated into the revised manuscript (replacing the current table with the updated version upon acceptance), clearly demonstrate the robustness and generalizability of our method.
>
> |Category|Metrics|*AlphaFold3*|*Phenix.real_space_refine*|*CryoNet.Refine*|
> |--|--|--|--|--|
> |**Model–map Correlation**|$\rm CC_{mask}\uparrow$|0.38|0.54|**0.59**|
> ||$\rm CC_{box}\uparrow$|0.41|0.53|**0.57**|
> ||$\rm CC_{mc}\uparrow$|0.40|0.55|**0.60**|
> ||$\rm CC_{sc}\uparrow$|0.39|0.55|**0.58**|
> ||$\rm CC_{peaks}\uparrow$|0.27|0.40|**0.45**|
> ||$\rm CC_{volume}\uparrow$|0.42|0.55|**0.60**|
> |**Model Geometric Metrics**|Angle RMSD (degree)$\downarrow$|1.58|0.72|**0.37**|
> ||$C_\beta$ deviations$\downarrow$|0.03|**0.00**|**0.00**|
> ||Ramachandran favored (%)$\uparrow$|95.73|96.39|**98.92**|
> ||Ramachandran outlier (%)$\downarrow$|0.82|**0.02**|0.06|
> ||Rotamer favored (%)$\uparrow$|97.08|85.42|**98.64**|
> ||Rotamer outlier (%)$\downarrow$|1.08|1.15|**0.49**|
>
> > Performance on protein complex (↑: higher is better, ↓: lower is better).
>
> ---
>
> ***Questions:***
>
> **Q1.** If my assumption in Weakness 1 is correct, that no density or geometry loss is computed during inference, would it be beneficial to incorporate these losses as a form of test-time refinement?
>
> **Response Q1:** Thank you for this insightful suggestion. In fact, our method precisely implements this strategy: we compute density and geometry losses and use them for test-time optimization across multiple recycles, which is the core of our adaptive refinement approach.
>
> **Q2.** Figure 2 includes a Pairformer module, but it is never discussed in the text. Is this a Transformer-based component where cross-attention occurs between atom and sequence embeddings?
>
> **Response Q2:** Yes, the Pairformer is a Transformer-based component that follows the AlphaFold3 (Abramson et al., Nature 2024) architecture, and its key function is to perform cross-attention between the atom and the sequence embeddings.

---

> ### Author Response · Authors · 2025-11-19
> **Response to Reviewer 9SBf (Weaknesses 1~3 )**
>
> ***Weaknesses:***
> **W1:** It is confusing what exactly happens during a training step versus inference. I assume Fig. 2 provides an overview of the training process. During inference, there seems to be no computation of density or geometry loss, and the model likely performs only a single pass through the Atom Encoder, Sequence Embedder, and Diffusion Module. Clarifying this distinction would help readers better understand the workflow and computational efficiency.
>
> **Response W1:** Thank you for pointing out the confusion regarding the workflow in Figures 1 and 2; we will revise them for better clarity in the camera ready version. Crucially, our method **does not include a separate inference process** as assumed; the entire procedure is a single, iterative **training-and-optimization loop** that directly produces the refined atomic structure. The workflow is as follows:
>
> 1. Input the atomic structure and density map.
> 2. The atomic structure enters the training state, sequentially passing through the frozen Atom Encoder, Sequence Embedder, and Pairformer module — finally the trainable Diffusion Module.
> 3. At each recycle, **density and geometry loss are calculated and backpropagated** through the diffusion module to update the atomic structure, until early-stopping is triggered or the preset maximum number of recycles is reached.
>
> Therefore, the final atomic structure is the result of this loss-driven, full-pass optimization, which explains why no single-pass "inference" stage is defined.
>
> **W2.** Several simple yet informative baselines are missing — for example, numerical optimization starting from the initial structure. Including such comparisons would better contextualize the claimed improvements.
>
> **Response W2:** We agree that including a simple numerical optimization baseline is valuable. Following your advice, we implemented a numerical optimization baseline where atomic structure coordinates are treated as learnable parameters and optimized via SGD with the same loss functions (density map + geometric restraints). This approach completely bypasses neural network components—no diffusion module or structural encoders—providing a pure gradient-descent baseline that directly updates atomic structure coordinates.
>
> > Method Comparison Table (N=27)
>
> |Category|Metrics|*AlphaFold3*|*Phenix. real_space_refine*|*Numerical Optimization*|*CryoNet.Refine*|
> |--|--|--|--|--|--|
> |**Model–map Correlation**|$\rm CC_{mask}\uparrow$|0.44|0.61|0.46|**0.65**|
> ||$\rm CC_{box}\uparrow$|0.44|0.55|0.47|**0.58**|
> ||$\rm CC_{mc}\uparrow$|0.45|0.62|0.46|**0.66**|
> ||$\rm CC_{sc}\uparrow$|0.44|0.61|0.45|**0.64**|
> ||$\rm CC_{peaks}\uparrow$|0.31|0.44|0.32|**0.47**|
> ||$\rm CC_{volume}\uparrow$|0.47|0.61|0.48|**0.65**|
> |**Model Geometric Metrics**|Angle RMSD (degree)$\downarrow$|1.60|0.73|**0.27**|0.54|
> ||$C_\beta$ deviations$\downarrow$|0.09|**0.00**|**0.00**|**0.00**|
> ||Ramachandran favored (%)$\uparrow$|95.48|96.33|98.03|**99.68**|
> ||Ramachandran outlier (%)$\downarrow$|1.04|**0.05**|0.22|0.11|
> ||Rotamer favored (%)$\uparrow$|96.8|80.99|97.43|**98.58**|
> ||Rotamer outlier (%)$\downarrow$|1.14|1.48|0.87|**0.51**|
>
> > Performance on protein complex (↑: higher is better, ↓: lower is better).
>
> The results above show that our CryoNet.Refine outperforms numerical optimization method both on model-map correlation coefficients and model geometric metrics, which highlights the critical role of our diffusion module (initialized with Boltz-2, a reproduction of AlphaFold3 (Abramson et al., Nature 2024). This module was pre-trained on over 270K macromolecular structures, learning rich priors about natural protein structures and the underlying rules, which is entirely absent in a simple numerical optimization scheme. It's one of the major factors why our method outperforms the baseline described above.

---

> > ### Comment · Reviewer_9SBf · 2025-11-25
> >
> > Thanks to the authors for responding. After reading the reply and also other reviewers’ opinions, I still have several further questions:
> >
> > 1. As regards my W3, I didn’t really see any discussion on that part in the rebuttal, so I’m still concerned about the ablation study here.
> >
> > 2. Regarding the density generator, thanks for the response and also for having a figure in Appendix D to show data patterns and visualizations. However, this still does not fully address my curiosity about how the density generator actually behaves. It would be good to provide more discussion or clarification, as also raised by reviewer yjst.

---

> > > ### Author Response · Authors · 2025-11-26
> > > **Response to Reviewer 9SBf**
> > >
> > > 1.As regards my W3, I didn't really see any discussion on that part in the rebuttal, so I'm still concerned about the ablation study here.
> > >
> > > **Response W3：** We have indeed conducted this ablation study, which is detailed in Appendix Fig. 11–12. Our results indicated that the typical 200-step diffusion sampler produces structures with noticeably lower model-map correlation coefficients (CC) and worse model geometric metrics. We suspect that the degradation arises from 2 main factors: (1) We employ a deterministic sampler, which intrinsically requires fewer steps; and (2) The diffusion module receives as input an already complete and physically reasonable macromolecular structure, rather than random noise as in the training regimes of AF3 or Boltz-2 for de novo structure prediction.
> > >   Furthermore, we have expanded the original ablation study to include a quantitative comparison between multi-step and our one-step approach across 27 cases. The results, as shown in the figure below, further demonstrate the rationale and effectiveness of the one-step strategy.
> > >
> > > |Category|Metrics|CryoNet.Refine|Classical Diffusion (200 steps)|
> > > |--|--|--|--|
> > > |**Model–map Correlation**|CC_mask ↑|0.65|0.20|
> > > ||CC_box ↑|0.58|0.28|
> > > ||CC_mc ↑|0.66|0.21|
> > > ||CC_sc ↑|0.63|0.21|
> > > ||CC_peaks ↑|0.47|0.13|
> > > ||CC_volume ↑|0.65|0.27|
> > > |**Model Geometric Metrics**|Angle RMSD (°) ↓|0.54|9.80|
> > > ||C-beta deviations ↓|0.00|33.04|
> > > ||Ramachandran favored (%) ↑|98.80|74.01|
> > > ||Ramachandran outlier (%) ↓|0.10|12.39|
> > > ||Rotamer favored (%) ↑|98.58|81.86|
> > > ||Rotamer outlier (%) ↓|0.51|9.21|
> > >
> > > ---
> > >
> > > 2. Regarding the density generator, thanks for the response and also for having a figure in Appendix D to show data patterns and visualizations. However, this still does not fully address my curiosity about how the density generator actually behaves. It would be good to provide more discussion or clarification, as also raised by reviewer yjst.
> > >
> > > **Response for density generator's function:**
> > >
> > > The density generator functions as a differentiable mapping that converts atomic structure coordinates into a synthetic density map. This synthetic map is then compared against the input experimental density map to compute a density loss during the training stage. This density loss acts as a guidance term that drives the diffusion module to iteratively adjust the atomic coordinates, ensuring that their resulting generated densities align with the experimental observations.
> > > The optimization thus couples structural refinement and density agreement in an end-to-end differentiable manner:
> > > $$L = \gamma_\text{den} L_\text{den} + L_\text{geo}.$$
> > > Through backpropagation of $L_\text{den}$, the diffusion module learns to modify local atom positions—tightening backbone traces and correcting side-chain placements—until the synthetic map becomes maximally correlated with $d_0$. This mechanism effectively enables the input experimental density map itself to act as a structural constraint guiding refinement.
> > > We will further clarify this process in the revision (Appendix D) .

---

### Meta-Review · Area_Chair_Y5VN · 2026-01-07

**Summary:**

CryoNet.Refine is a well-motivated refinement framework that bridges modern folding priors with experimental cryo-EM densities. It integrates a one-step diffusion sampler with differentiable density and geometry losses and performs per-structure optimization. Reviewers agree it consistently improves map–model correlation and stereochemistry over Phenix on the reported benchmarks, and it meaningfully reduces manual effort. The rebuttal clarifies training vs inference (recycling with test-time losses), explains the Pairformer, and expands experiments from 53 to 110 complexes. It adds a direct coordinate-optimization baseline and a more thorough diffusion ablation, both favoring CryoNet.Refine. These additions strengthen the empirical case, improve clarity, and support the claim that harnessing learned structural priors plus density/geometry guidance yields better refinement than classical optimization alone.

**Reviewer Concerns:**

he rebuttal addresses several key points. It clarifies the workflow: “one-step” refers to the deterministic sampler within each recycle, while refinement proceeds via multiple recycles with density and geometry losses computed at test time. It confirms per-structure optimization, details the Pairformer’s role, and reports expanded evaluations to 110 complexes. The added numerical-optimization baseline directly answers requests for a simple comparator and shows that learned priors materially help beyond pure gradient descent. The diffusion ablation now quantifies accuracy differences to multi-step sampling and supports the chosen design. The density generator is described as a differentiable Gaussian scattering simulator implemented in PyTorch, and the authors report map–map correlations versus experimental densities.

Some issues remain, but are not fatal for a poster. The density generator’s novelty mainly lies in differentiability rather than new physics; the paper should sharpen that claim and include quantitative comparisons to standard simulators. Methodological specifics (loss weights, runtime profiles, recycle sensitivity) can be expanded in the camera-ready. Efficiency is still limited by per-structure optimization, but the authors outline plausible acceleration paths. Overall, the rebuttal meaningfully reduces uncertainty on correctness and practical value, and the results are strong for a refinement task across diverse complexes.

**Reviewer Scores:**

Reviewer 1 (initial 6): Likely unchanged. The workflow clarifications and added results address several points, but remaining concerns about missing baselines (historically broader set), density generator analysis, and scale persist.

Reviewer 2 (initial 6): Likely unchanged or slightly up. Their question on direct coordinate optimization was answered with a baseline and favorable results. Efficiency remains a concern, but they may stay at 6.

Reviewer 3 (initial 6): Likely unchanged. Clarifications on one-step vs recycling help, and per-structure training is answered. However, differences from AF3 and the exact architectural details still need tighter exposition. Score likely stays at 6.

Reviewer 4 (initial 4): Possibly up to 5. Their key ask for a coordinate-optimization baseline and quantitative diffusion ablation was addressed with positive outcomes. But questions about the novelty of the density generator and necessity of diffusion persist, so full reversal is unlikely.

---

### Decision · Program_Chairs · 2026-01-26

Accept (Poster)